# Hierarchical World Models as Visual Whole-Body Humanoid Controllers

**Nicklas Hansen**[1]   **Jyothir S V**[2]   **Vlad Sobal**[2]   **Yann LeCun**[2,3]
**Xiaolong Wang**[1*]   **Hao Su**[1*]

[1]UC San Diego   [2]New York University   [3]Meta AI

[*]Equal advising

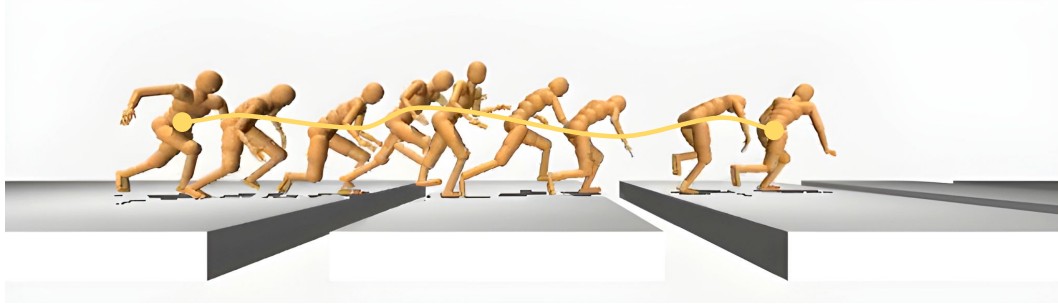

*Figure 1.* **Visual whole-body control for humanoids.** We present `Puppeteer`, a hierarchical world model for humanoid control with visual observations. Our method produces natural and human-like motions *without* any reward design or skill primitives, and traverses challenging terrain.

## Abstract

Whole-body control for humanoids is challenging due to the high-dimensional nature of the problem, coupled with the inherent instability of a bipedal morphology. Learning from visual observations further exacerbates this difficulty. In this work, we explore highly data-driven approaches to visual whole-body humanoid control based on reinforcement learning, without any simplifying assumptions, reward design, or skill primitives. Specifically, we propose a hierarchical world model in which a high-level agent generates commands based on visual observations for a low-level agent to execute, both of which are trained with rewards. Our approach produces highly performant control policies in 8 tasks with a simulated 56-DoF humanoid, while synthesizing motions that are broadly preferred by humans.

**Code and videos:** **https://www.nicklashansen.com/rlpuppeteer**

## 1 Introduction

Learning a generalist agent in the physical world is a long-term goal of many researchers in AI. Among variant agent designs, humanoids stand out as versatile platforms capable of performing a wide range of tasks, by integrating whole-body control and perception. However, this is a very challenging problem due to the high-dimensional nature of the observation and action spaces, as well as the complex dynamics of a bipedal embodiment, and it makes learning successful yet natural whole-body controllers with reinforcement learning (RL) extremely difficult. For example, consider the task shown in Figure 1, where a humanoid is rewarded for forward progress while jumping over gaps. To succeed in this task, a humanoid needs to accurately perceive the position and length of oncoming floor gaps, while carefully coordinating full body motions such that it has sufficient momentum and range to reach across each gap.

Due to the sheer complexity of such problems, prior work choose to make simplifying assumptions, such as using low-dimensional (privileged) observations and actions (Heess et al., 2017; Peng et al., 2018; Wagener et al., 2022; Jiang et al., 2023; Peng et al., 2021), or (learned) skill primitives (Merel et al., 2017; 2018b; Hasenclever et al., 2020; Peng et al., 2022; Cheng et al., 2024). Most related

to our work, MoCapAct (Wagener et al., 2022) first learn ~2600 individual tracking policies via RL, then distill them into a multi-clip tracking policy via imitation learning, and subsequently train a high-level RL policy to output goal embeddings for the multi-clip policy to track. While such approaches have been shown to transfer to simple reaching and velocity control tasks from proprioceptive inputs, we expect to find a solution that can perform *complex, visual whole-body control tasks* while remaining entirely data-driven and relying on as few assumptions as possible. In this paper, we propose to directly learn a visual controller for high-dimensional humanoid robot joints via model-based RL and an existing large-scale motion capture (MoCap) dataset (CMU, 2003), while requiring *several orders of magnitude* less interactions to learn new tasks compared to prior work.

We propose a data-driven RL method for visual whole-body control that produces natural, human-like motions and can perform diverse tasks. Our approach, dubbed `Puppeteer`, is a hierarchical JEPA-style (LeCun, 2022) world model that consists of two distinct agents: a proprioceptive *tracking* agent that tracks a reference motion via joint-level control, and a visual *puppeteer* agent that learns to perform downstream tasks by synthesizing lower-dimensional reference motions for the tracking agent to track based on visual observations.

Concretely, the two agents are trained independently in two separate stages using the model-based RL algorithm TD-MPC2 (Hansen et al., 2024) as a learning backbone. First, a *single* tracking world model is (pre)trained to track reference motions from pre-existing human MoCap data (CMU, 2003) re-targeted to a humanoid embodiment (Wagener et al., 2022). It learns a single model to convert any reference kinematic motion to physically executable actions. This is a departure from previous work that learns multiple low-level models (Merel et al., 2017; Hasenclever et al., 2020; Wagener et al., 2022). Importantly, this tracking agent can be saved and reused across all downstream tasks. In the second stage, we train a puppeteering world model that takes visual observation as inputs and outputs the reference motion for the tracking agent based on the specified downstream task. The puppeteer agent is trained with online environment interaction using the fixed tracking agent. A key feature of our framework is its striking *simplicity*: both world models are algorithmically identical (but differ in inputs/outputs) and can be trained using RL *without any bells and whistles*. Different from a traditional hierarchical RL setting, our puppeteer agent (high-level policy) outputs geometric locations for a small number of end-effector joints instead of a goal embedding. The tracking agent (low-level policy) is thus only required to learn joint-level physics. This makes the tracking agent easily sharable and generalizable across tasks, leading to an overall small computational footprint.

To evaluate the efficacy of our approach, we propose a new task suite for visual whole-body humanoid control with a simulated 56-DoF humanoid, which contains a total of 8 challenging tasks. We show that our method produces highly performant control policies across all tasks compared to a set of strong model-free and model-based baselines: SAC (Haarnoja et al., 2018), DreamerV3 (Hafner et al., 2023), and TD-MPC2 (Hansen et al., 2024). Furthermore, we find that motions generated by our method are broadly preferred by humans in a user study with 51 participants. We conclude the paper by carefully dissecting how each of our design choices influence results. Code for method and environments is available at **https://www.nicklashansen.com/rlpuppeteer**. Our main contributions can be summarized as follows:

- **Task suite.** We propose a new, challenging task suite for visual whole-body humanoid control with a simulated 56-DoF humanoid. The task suite has 8 tasks in total, and poses a significant challenge for existing state-of-the-art RL algorithms. At present, no such benchmark exists.
- **Hierarchical world model.** We propose a simple yet highly effective method for high-dimensional continuous control that uses a learned hierarchical world model for planning.
- **Evaluating "naturalness" of controllers.** We develop several metrics for quantifying how natural and human-like generated motions are across tasks in our suite, including human preferences from a user study. To the best of our knowledge, no prior work has explicitly evaluated naturalness of learned policies for humanoid control.
- **Analysis & ablations.** We carefully ablate each of our design choices, analyze the relative importance of each component, and provide actionable advice for future work in this area.

## 2  PRELIMINARIES

**Problem formulation.** We model visual whole-body humanoid control as a reinforcement learning problem governed by an episodic Markov Decision Process (MDP) characterized by the tuple

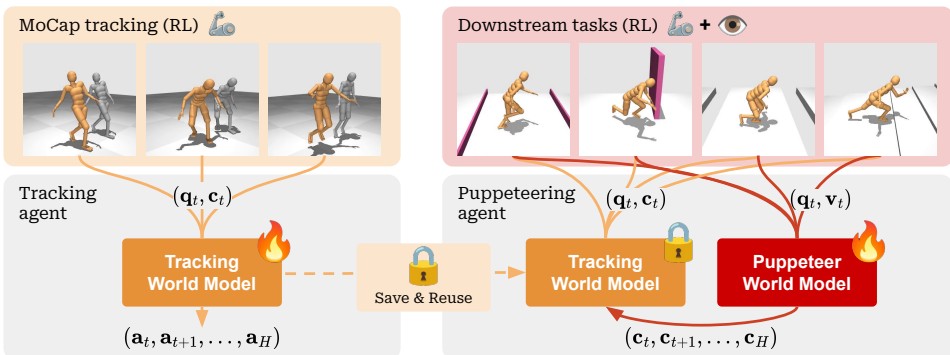

*Figure 2.* **Approach.** We pretrain a tracking agent (world model) on human MoCap data using RL; this agent takes proprioceptive information $\mathbf{q}_t$ and an abstract reference motion (command) $\mathbf{c}_t$ as input, and synthesizes $H$ low-level actions that tracks the reference motion. We then train a high-level puppeteering agent on downstream tasks via online interaction; this agent takes both state $\mathbf{q}_t$ and visual information $\mathbf{v}_t$ as input, and outputs commands for the tracking agent to execute.

$(\mathcal{S}, \mathcal{A}, \mathcal{T}, R, \gamma, \Delta)$ where $\mathbf{s} \in \mathcal{S}$ are states, $\mathbf{a} \in \mathcal{A}$ are actions, $\mathcal{S}: \mathcal{S} \times \mathcal{A} \mapsto \mathcal{S}$ is the environment transition (dynamics) function, $R: \mathcal{S} \times \mathcal{A} \mapsto \mathbb{R}$ is a scalar reward function, $\gamma$ is the discount factor, and $\Delta: \mathcal{S} \mapsto \{0, 1\}$ is an episode termination condition. We implicitly consider both proprioceptive information $\mathbf{q}$ and visual information $\mathbf{v}$ as part of states $\mathbf{s}$ and will only make the distinction clear when necessary. We aim to learn a policy $\pi: \mathcal{S} \mapsto \mathcal{A}$ that maximizes discounted sum of rewards in expectation: $\mathbb{E}_\pi \left[ \sum_{t=0}^{T} \gamma^t r_t \right]$, $r_t = R(\mathbf{s}_t, \pi(\mathbf{s}_t))$ for an episode of length $T$, while synthesizing motions that look *"natural"*. We informally define natural motions as policy rollouts that are human-like, but develop several metrics for measuring the "naturalness" of policies in Section 4.

**TD-MPC2.** We build upon the model-based reinforcement learning (MBRL) algorithm TD-MPC2 (Hansen et al., 2024), which represents the state-of-the-art in continuous control and has been shown to outperform alternatives in tasks with high-dimensional action spaces (Hansen et al., 2022; 2024; Sferrazza et al., 2024). Specifically, TD-MPC2 learns a latent decoder-free world model from environment interactions and selects actions by planning with the learned model. All components of the world model are learned end-to-end using a combination of joint-embedding prediction (Grill et al., 2020), reward prediction, and temporal difference (Sutton, 1998) losses, *without* decoding raw observations. During inference, TD-MPC2 follows the Model Predictive Control (MPC) framework for local trajectory optimization using Model Predictive Path Integral (MPPI) (Williams et al., 2015) as a derivative-free (sampling-based) optimizer. To accelerate planning, TD-MPC2 additionally learns a model-free policy prior which is used to warm-start the sampling procedure.

## 3 A HIERARCHICAL WORLD MODEL FOR HIGH-DIMENSIONAL CONTROL

We aim to learn highly performant and *"natural"* policies for visual whole-body humanoid control in a data-driven manner using hierarchical world models. A key strength of our approach is that it can synthesize human-like motions without any explicit domain knowledge, reward design, nor skill primitives. While we focus on humanoid control due to their complexity, our approach can in principle be applied to any embodiment. Our method, dubbed `Puppeteer`, consists of two distinct agents, both of which are implemented as TD-MPC2 world models (Hansen et al., 2024) and trained independently. Figure 2 provides an overview of our method. The two agents are designed as follows:

1. A low-level *tracking* agent that takes a robot proprioceptive state $\mathbf{q}_t$ and an abstract command $\mathbf{c}_t$ as input at time $t$, and uses planning with a learned world model to synthesize a sequence of $H$ control actions $\{\mathbf{a_t}, \mathbf{a_{t+1}}, \dots, \mathbf{a_{t+H}}\}$ that (approximately) obeys the abstract command.

2. A high-level *puppeteering* agent that takes the same robot proprioceptive state $\mathbf{q}_t$ as input, as well as (optionally) auxiliary information and modalities such as RGB images $\mathbf{v}_t$ or task-relevant information, and uses planning with a learned world model to synthesize a sequence of $H$ high-level abstract commands $\{\mathbf{c_t}, \mathbf{c_{t+1}}, \dots, \mathbf{c_{t+H}}\}$ for the low-level agent to execute.

A unique benefit of our approach is that *a single tracking world model can be (pre)trained and reused across all downstream tasks*. This is in contrast to much of prior work that either learn a large number (up to ∼2600) of low-level policies (Merel et al., 2017; 2018b; Hasenclever et al., 2020; Wagener et al., 2022), or train policies from scratch on each downstream task (Peng et al., 2018; 2021). The tracking and puppeteering world models are algorithmically identical (but differ in inputs/outputs), and consist of the following 6 components:

$$
\begin{array}{llll}
\text{Encoder} & \mathbf{z} = h(\mathbf{s}) & \triangleright \text{ Encodes state into a latent embedding} \\
\text{Latent dynamics} & \mathbf{z}' = d(\mathbf{z}, \mathbf{a}) & \triangleright \text{ Predicts next latent state} \\
\text{Reward} & \hat{r} = R(\mathbf{z}, \mathbf{a}) & \triangleright \text{ Predicts reward } r \text{ of a state transition} \\
\textbf{Termination} & \hat{\delta} = D(\mathbf{z}, \mathbf{a}) & \triangleright \textbf{ Predicts probability of termination} \\
\text{Terminal value} & \hat{q} = Q(\mathbf{z}, \mathbf{a}) & \triangleright \text{ Predicts discounted sum of rewards} \\
\text{Policy prior} & \hat{\mathbf{a}} = p(\mathbf{z}) & \triangleright \text{ Predicts an action } \mathbf{a}^* \text{ that maximizes } Q
\end{array}
\tag{1}
$$

where $\mathbf{z}$ is a latent state. Because we consider episodic MDPs with termination conditions, we additionally add a termination prediction head $D$ (**highlighted** in Equation 1) that predicts the probability of termination conditioned on a latent state and action. Use of termination signals in the context of planning with a world model requires special care and has, to the best of our knowledge, not been explored in prior work; we introduce a novel method for this in Section 3.3. In the following, we describe the two agents and their interplay in the context of visual whole-body humanoid control.

## 3.1 LOW-LEVEL TRACKING WORLD MODEL

We first train the low-level tracking world model independently from the high-level agent and any potential downstream tasks. We leverage pre-existing human MoCap data (CMU, 2003) retargeted to the 56-DoF "CMU Humanoid" embodiment (Tassa et al., 2018) during training of the tracking model, which (as we will later show empirically) implicitly encodes human motion priors. Specifically, we train our tracking world model by sampling $(\mathbf{s}_t, \mathbf{a}_t, r_t, \mathbf{s}_{t+1}, \ldots, r_H)$ sequences from MoCapAct (Wagener et al., 2022), an offline dataset that consists of noisy, suboptimal rollouts from existing policies trained to track reference motions (836 MoCap clips). This is in contrast to prior literature that learn per-clip policies or skill primitives (Heess et al., 2017; Merel et al., 2017; Hasenclever et al., 2020).

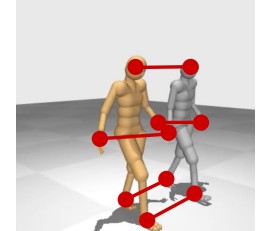

*Figure 3.* **MoCap tracking.** The low-level tracking agent is trained to track relative end-effector (head, hands, feet) positions of sampled reference motions in 3D space.

Observations include humanoid proprioceptive information $\mathbf{q}_t$ at time $t$, as well as a reference motion (command) $\mathbf{c}_t$ to track. During training of the tracking policy, we let $\mathbf{c}_t \doteq (\mathbf{q}_{t+1\ldots t+H}^{\text{ref}})$ where each $\mathbf{q}^{\text{ref}}$ corresponds to relative end-effector (head, hands, feet) positions of the sampled reference motion at a future timestep; during downstream tasks, we train the high-level agent to output (via planning) commands $\mathbf{c}$ for the low-level agent to track. Figure 3 illustrates our low-dimensional reference; the **controllable humanoid** tracks **end-effector positions** of a **reference motion**. We label all transitions using the reward function from Hasenclever et al. (2020). To improve state-action coverage of the tracking world model, we train with a combination of offline data and online interactions, maintaining a separate replay buffer for online interaction data and sampling offline/online data with a 50%/50% ratio in each gradient update as in Feng et al. (2023). We find this to be crucial for tracking performance when training a single world model on a large number of MoCap clips.

## 3.2 HIGH-LEVEL PUPPETEERING WORLD MODEL

We now consider training a high-level puppeteering world model via online interaction in downstream tasks. As illustrated in Figure 2, the puppeteering model is trained (using downstream task rewards) to control the tracking model via commands $\mathbf{c}$, *i.e.*, we redefine commands to now be the action space of the puppeteering agent. The tracking world model remains frozen (no weight updates) throughout this process, which allows us to reuse the *same* tracking model across *all* downstream tasks. Because the high-level agent uses planning for action selection, it natively supports temporal abstraction by outputting multiple commands $(\mathbf{c}_t, \mathbf{c}_{t+1}, \ldots, \mathbf{c}_{t+H})$ for the low-level agent

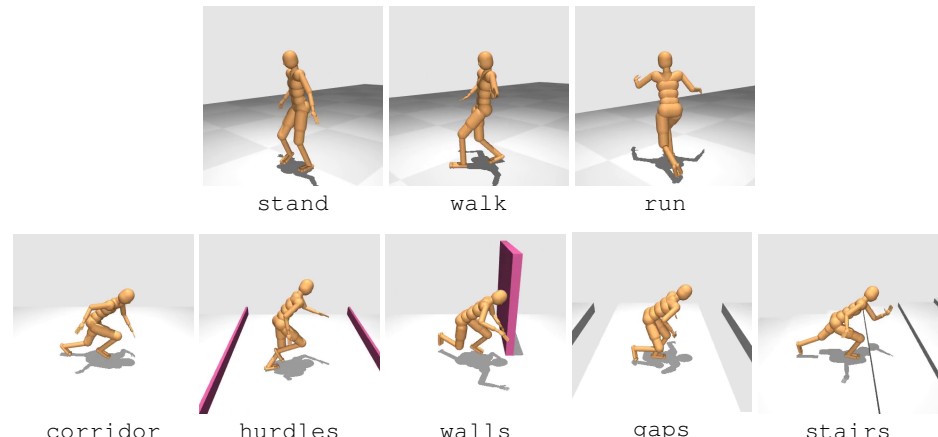

*Figure 4.* **Tasks.** We develop 5 visual whole-body humanoid control tasks with a 56-DoF simulated humanoid (bottom), as well as 3 non-visual tasks (top). See Appendix E for more details.

to execute; we treat the number of low-level steps per high-level step as a hyperparameter $k$ that allows us to trade strong motion prior (large $k$) for control granularity (small $k$). The high-level policy outputs commands at a fixed frequency regardless of whether the previous command was achieved.

### 3.3 PLANNING WITH TERMINATION CONDITIONS

We consider episodic MDPs with termination conditions. In the context of humanoid control, a common such termination condition is non-foot contact with the floor. Use of termination conditions requires special care in the context of world model learning and planning, as both components are used to simulate (latent) multi-step rollouts. We extend the world model of TD-MPC2 with a termination prediction head $D$, which predicts the probability of termination at each time step. This termination head is trained end-to-end together with all other components of the world model using

$$\mathcal{L}_{\text{Puppeteer}}(\theta) \doteq \mathcal{L}_{\text{TD-MPC2}}(\theta) + \alpha \, \text{CE}(\hat{\delta}, \delta) \tag{2}$$

where $\hat{\delta}, \delta$ are predicted and ground-truth termination signals, respectively, CE is the (binary) cross-entropy loss, and $\alpha$ is a constant coefficient balancing the losses. We additionally truncate TD-targets at terminal states during training. It is similarly necessary to truncate model rollouts and value estimates during planning (at test-time). However, we only have access to predicted termination signals at test-time, which can be noisy and consequently lead to high-variance value estimates for latent rollouts. To mitigate this, we maintain a cumulative weighting (discount) of termination probabilities when rolling out the model (capped at $0$), such that only a *soft* truncation is applied.

## 4 EXPERIMENTS

Our proposed method holds the promise of strong downstream task performance while still synthesizing natural and human-like motions. To evaluate the efficacy of our method, we propose a new task suite for whole-body humanoid control with multi-modal observations (vision and proprioceptive information) based on the "CMU Humanoid" model from DMControl (Tassa et al., 2018). Our simulated humanoid has 56 fully controllable joints ($\mathcal{A} \in \mathbb{R}^{56}$), and includes both head, hands, and feet. We aim to learn highly performant policies in a data-driven manner without the need for embodiment- or task-specific engineering (*e.g.*, reward design, constraints, or auxiliary objectives), while synthesizing natural and human-like motions. Code for method and environments is available at https://www.nicklashansen.com/rlpuppeteer.

### 4.1 EXPERIMENTAL DETAILS

**Tasks.** Our proposed task suite consists of 5 vision-conditioned whole-body locomotion tasks, and an additional 3 tasks without visual input. We provide an overview of tasks in Figure 4; they are designed with a high degree of randomization and include running along a corridor, jumping over

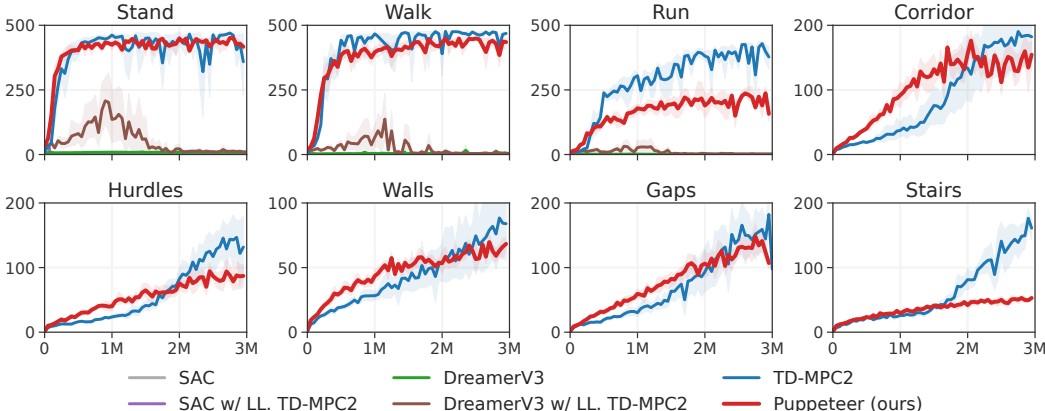

*Figure 5.* **Learning curves.** Episode return vs. environment steps on all 8 tasks from our proposed task suite. Our method generally matches the return of TD-MPC2 on these tasks while producing more natural motions. We only evaluate SAC and DreamerV3 on proprioceptive tasks as they do not achieve any meaningful performance. Average of 10 random seeds; shaded area is $95\%$ CIs.

hurdles and gaps, walking up the stairs, and circumnavigating obstacles (walls). All 5 visual control tasks use a reward function that is proportional to the linear forward velocity, while non-visual tasks reward displacement in any direction. Episodes are terminated at timeout (500 steps) or when a non-foot joint makes contact with the floor. We empirically observe that the TD-MPC2 baseline degenerates to highly unrealistic behavior without a contact-based termination condition, and thus modify TD-MPC2 to support termination as described in Section 3.3. See Appendix E for details.

**Implementation.** We pretrain a single 5M parameter TD-MPC2 world model to track all 836 CMU MoCap (CMU, 2003) reference motions retargeted to the CMU Humanoid model. This in contrast to, *e.g.*, MoCapAct (Wagener et al., 2022) that trains ∼2600 individual tracking policies. Our tracking agent is trained for 10M steps using both offline data (noisy rollouts) from MoCapAct (Wagener et al., 2022) and online interaction with a new reference motion sampled in each episode. We sample $50\%$ of each batch from the offline dataset, and $50\%$ from the online replay buffer for each gradient update; we did not experiment with other ratios. The puppeteering agent is similarly implemented as a 5M parameter TD-MPC2 world model, which we train from scratch via online interaction on each downstream task. Observations include a 212-d proprioceptive state vector and $64 \times 64$ RGB images from a third-person camera. Both agents act at the same frequency, *i.e.*, we set $k = 1$. Training the tracking world model takes approximately 12 days, and training the puppeteering world model takes approximately 4 days, both on a single NVIDIA GeForce RTX 3090 GPU. CPU and RAM usage is negligible. System requirements are detailed in Appendix C.

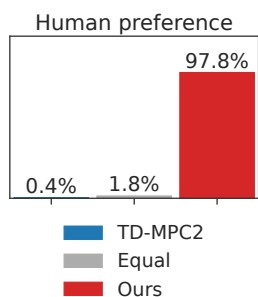

*Figure 6.* **Human preference in humanoid motions.** Aggregate results from a user study ($n = 51$) where humans are presented with pairs of motions generated by TD-MPC2 and our method, and are asked to provide their preference.

**Baselines.** We benchmark our method against state-of-the-art RL algorithms for continuous control, including *(1)* widely used model-free RL method **Soft Actor-Critic** (SAC) (Haarnoja et al., 2018) which learns a stochastic policy and value function using a maximum entropy RL objective, *(2)* model-based RL method **DreamerV3** (Hafner et al., 2020; 2021; 2023) which simultaneously learns a world model using a generative objective, and a model-free policy in the latent space of the learned world model, *(3)* model-based RL method **TD-MPC2** (Hansen et al., 2022; 2024) which learns a self-predictive (decoder-free) world model and selects actions by planning with the learned world model, *(4)* a hierarchical baseline that uses the same low-level TD-MPC2 agent as our method but trains a SAC policy as the high-level agent, and (5) the same hierarchical baseline but with DreamerV3 as the high-level agent (and TD-MPC2 at the low level). We refrain from making a direct comparison to MoCapAct (Wagener et al., 2022) and DeepMimic (Peng et al., 2018) as they do not support visual observations and require several orders of magnitude more environment interactions to learn downstream tasks. Both our method and baselines use

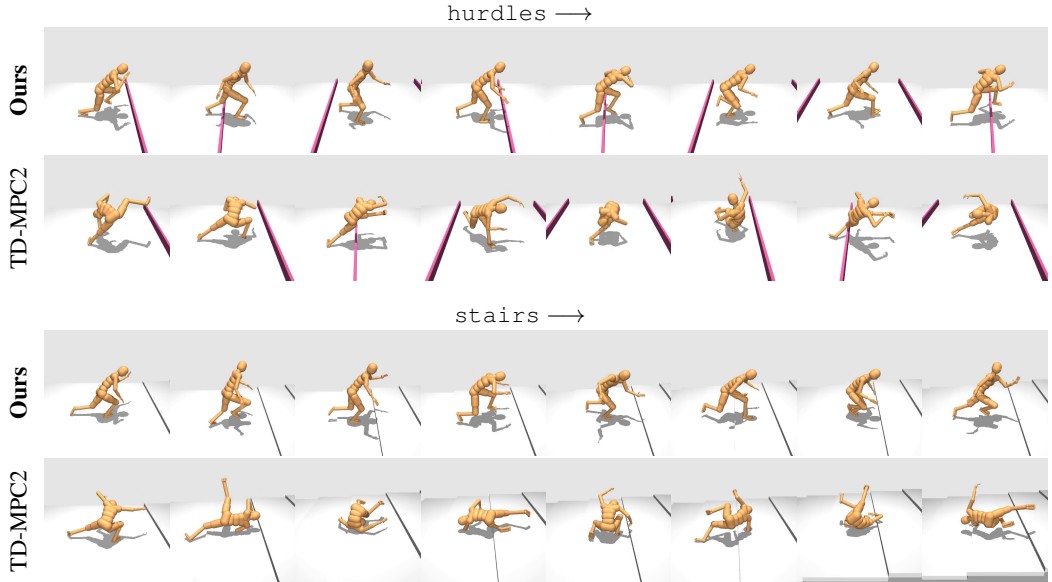

*Figure 7.* **Qualitative results.** Our hierarchical approach, `Puppeteer`, produces natural human motions, whereas TD-MPC2 trained end-to-end often learns high-performing but unnatural gaits.

the **same** hyperparameters across all tasks, as TD-MPC2 and DreamerV3 have been shown to be robust to hyperparameters across task suites (Hansen et al., 2024; Hafner et al., 2023; Sferrazza et al., 2024). For a fair comparison, we experiment with various design choices and hyperparameter configurations for SAC and report the best results that we obtained. We provide further implementation details in Appendix D.

## 4.2 MAIN RESULTS

We first present our main benchmark results, and then analyze and ablate each design choice.

**Benchmark results.** We evaluate our method, `Puppeteer`, and baselines on all 8 whole-body humanoid control tasks. Episode return as a function of environment steps is shown in Figure 5. We observe that the performance of our method is comparable to that of TD-MPC2 across all tasks (except *stairs*), whereas SAC and DreamerV3 does not achieve any meaningful performance within our computational budget of 3M environment steps; hierarchical DreamerV3 achieves non-trivial yet still poor performance with TD-MPC2 as the low-level agent. As we will soon reveal, TD-MPC2 produces better policies in terms of episode return on the *stairs* task, but far less natural behavior (walking vs. rolling up stairs). We conjecture that this is a symptom of *reward hacking* (Clark & Amodei, 2016; Skalse et al., 2022). Sample videos are available at https://www.nicklashansen.com/rlpuppeteer.

**"Naturalness" of humanoid controllers.** We conduct a user study ($n = 51$) in which humans are shown pairs of short ($\sim$10s) clips of policy rollouts from TD-MPC2 and our method, and are asked to provide their preference. Participants are undergraduate and graduate students across multiple universities and disciplines. Results from this study are shown in Figure 6, and Figure 7 shows two sample clips from the study. While both methods perform comparably in terms of downstream task reward, a supermajority of participants rate rollouts from our method as more natural than that of TD-MPC2, with only 4% of responses rating them as "equally natural" and 0% rating TD-MPC2 as more natural. This preference is especially pronounced in the *stairs* task, where TD-MPC2 achieves a higher asymptotic return (higher forward velocity) but learns to "roll" up stairs as opposed to our method that walks. We also report several quantitative measures of naturalness in Table 1, which strongly support our user

*Table 1.* **Proxies for "naturalness".** Evaluated on the *gaps* task. *eplen* denotes the average episode length (survival time) at 1M steps and at convergence; *height* is the average torso height (gait) at convergence. Mean and std. across 10 seeds.

| | eplen@1M ↑ | eplen ↑ | height (cm) ↑ |
|---|---|---|---|
| TD-MPC2 | $66.9 \pm 9.8$ | $181.6 \pm 28.1$ | $85.9 \pm 4.7$ |
| Ours | $\mathbf{115.9 \pm 5.2}$ | $159.3 \pm 5.9$ | $\mathbf{96.0 \pm 0.2}$ |

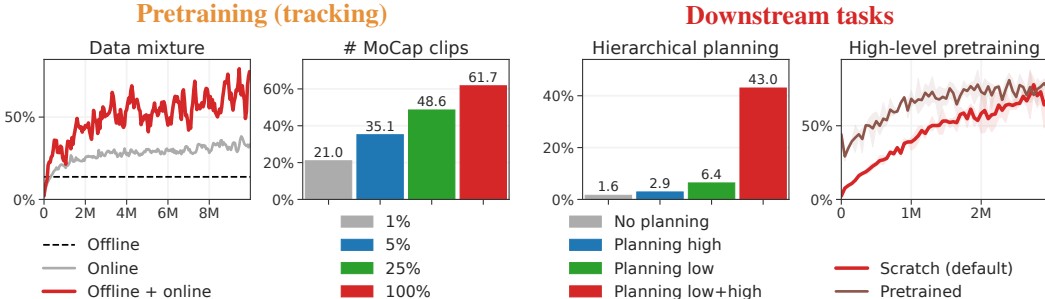

*Figure 8.* **Ablations.** Normalized score for various ablations of `Puppeteer` during pretraining (*left*) and downstream tasks (*right*). Pretraining benefits from diverse data, as well as both pre-existing (offline) data and online interactions. We also observe that planning is critical to whole-body humanoid control. Mean across 3 seeds; downstream ablations are averaged across 5 tasks.

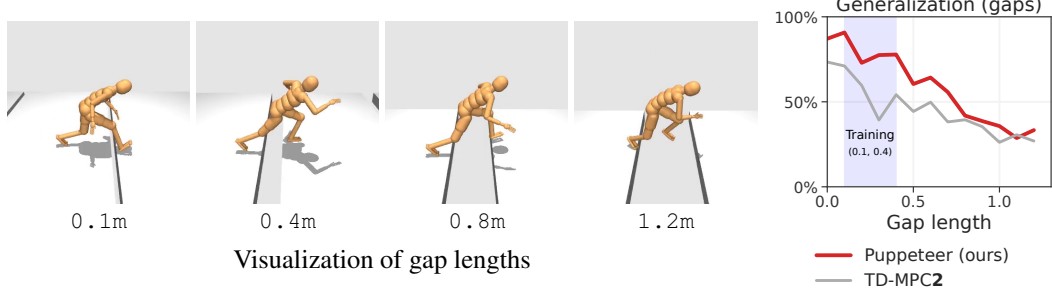

*Figure 9.* **Zero-shot generalization to larger gap lengths.** *(Left)* Visualization of gap lengths. Agent is trained on gaps $[0.1, 0.4]$m and evaluated on gaps up to $1.2$m. *(Right)* Normalized performance as a function of gap length in the visual *gaps* task. Mean of 3 seeds. Our method achieves non-trivial performance on gaps up to $3\times$ the training data. CIs omitted for visual clarity.

study results. These findings underline the importance of a more holistic evaluation of RL policies as opposed to solely relying on rewards. See Appendix A for more results.

## 4.3 ABLATIONS & ANALYSIS

We ablate each design choice of our method, including both the pretraining (tracking) and downstream task learning stages. Our experimental results are summarized in Figure 8.

**Pretraining (tracking).** Our method leverages both offline and online data during pretraining of the tracking world model. We ablate this training mixture in two distinct ways: *(i)* using only offline or online data, and *(ii)* reducing the number of unique MoCap clips seen during training. Interestingly, we find that leveraging both data sources leads to better tracking policies overall. We hypothesize that this is because offline data may help in learning to track especially difficult motions such as jumping and balancing on one leg, while online data improves state-action coverage and thus leads to a more robust world model overall. Similarly, training on more diverse MoCap clips also leads to better tracking performance. Training on all 836 MoCap clips results in the best tracking world model, and we expect tracking to further improve with availability of more MoCap data. See Appendix B for more results.

**Downstream tasks.** We conduct three ablations that help us better understand the impact of a hierarchical approach to downstream tasks: *(i)* using a learned model-free policy in lieu of planning in either level of the hierarchy, *(ii)* pretraining of the *high-level* agent in addition to the low-level agent, and *(iii)* evaluating zero-shot generalization to unseen environment variations (gap length in the *gaps* task). The first two ablations are shown in Figure 8, and the latter is shown in Figure 9. We find that planning at both levels is critical to effective whole-body humanoid control, which we conjecture is due to the high dimensionality of the problem; this is supported by concurrent studies on high-dimensional continuous control with TD-MPC2 (Hansen et al., 2024; Sferrazza et al., 2024). Next, we pretrain agents on the *corridor* task and independently finetune on each visual control task. While the specific environments and motions differ between tasks, we find that our method benefits substantially from finetuning. We conjecture that this is because the need to control a low-level

tracking agent is shared between all high-level agents, irrespective of the downstream task. This is in contrast to contemporary work on pretraining that often pretrains on large-scale out-of-domain data (Hansen et al., 2023b; Xu et al., 2023). Finally, we explore the zero-shot generalization ability of our method to harder, unseen variations of the *gap* task. Interestingly, we observe that our method generalizes to gap lengths up to $3\times$ the training data without additional training. In light of these results, we believe that further investigation of the generalization ability of hierarchical world models will be a promising direction for future research.

## 5 RELATED WORK

**Learning whole-body controllers for humanoids** is a long-standing problem at the intersection of the machine learning and robotics communities. Humanoids are of particular interest to the learning community because of the high-dimensional nature of the problem (Heess et al., 2017; Merel et al., 2017; Peng et al., 2018; Merel et al., 2018b; Hasenclever et al., 2020; Won et al., 2022; Wagener et al., 2022; Shi et al., 2022; Caggiano et al., 2022; Sferrazza et al., 2024; He et al., 2024; Cheng et al., 2024), and to the robotics community because it is a promising morphology for general-purpose robotic agents (Grizzle et al., 2009; Li et al., 2023; BostonDynamics, 2024; Unitree, 2024; Cheng et al., 2024). Prior work predominantly focus on learning control policies for individual tasks using model-free reinforcement learning algorithms, with human MoCap data (CMU, 2003) incorporated via either adversarial reward terms (Peng et al., 2018; 2021; 2022) or learned skill encoders (Heess et al., 2017; Merel et al., 2018b; Hasenclever et al., 2020; Shi et al., 2022; Wagener et al., 2022). While adversarial reward terms can produce natural behavior, this class of methods suffer from poor sample-efficiency as they learn a control policy from scratch for each downstream task. Our work is most similar to the latter class of methods, which enables reuse of the low-level policy and/or skill encoder across tasks. Most related to ours, MoCapAct (Wagener et al., 2022) first learn ~2600 individual tracking policies via RL, then distill them into a multi-clip tracking policy via imitation learning, and subsequently train a high-level RL policy to output goal embeddings for the multi-clip policy to track. Their resulting representation is used to perform simple reaching and velocity control tasks from privileged state information in approx. 150M environment steps. Our method trains a *single* world model to track the entire MoCap dataset, and is reused to learn a variety of *visual* whole-body control tasks in $\leq$ 3M environment steps. Concurrent to our work, Humanoid-Bench (Sferrazza et al., 2024) similarly introduce a whole-body control benchmark using the less expressive Unitree H1 (Unitree, 2024) embodiment. Our contributions differ in two important ways: *(1)* we develop a method for synthesizing natural human motions with a highly expressive humanoid model while Sferrazza et al. (2024) benchmark existing methods for online RL without regard for naturalness, and *(2)* HumanoidBench solely considers tasks with privileged state information in their experiments (*i.e.*, no visual observations).

**World models** (and model-based RL more broadly) are of increasing interest to researchers due to their strong empirical performance in an online RL setting (Ha & Schmidhuber, 2018; Hafner et al., 2023; Hansen et al., 2024), as well as their promise of generalization to structurally similar problem instances (Zhang et al., 2018; Zheng et al., 2022; Lee et al., 2022; Xu et al., 2023; LeCun, 2022; Sobal et al., 2022; Brohan et al., 2023). Existing model-based RL algorithms can broadly be categorized into algorithms that select actions by planning with a learned world model (Ebert et al., 2018; Schrittwieser et al., 2020; Ye et al., 2021; SV et al., 2023; Hansen et al., 2024), and algorithms that instead learn a model-free policy using imagined rollouts from the world model (Kaiser et al., 2020; Hafner et al., 2023). We build upon the TD-MPC2 (Hansen et al., 2024) world model, which uses planning and has been shown to outperform existing algorithms for continuous control (Hansen et al., 2023a; Lancaster et al., 2023; Feng et al., 2023; Sferrazza et al., 2024). We demonstrate that planning is key to success in the high-dimensional continuous control problems that we consider.

**Hierarchical RL** offers a framework for subdividing a complex learning problem into more approachable subproblems, often by, *e.g.*, leveraging (learned or manually designed) skill primitives (Pastor et al., 2009; Merel et al., 2017; 2018a; Shi et al., 2022) or facilitating learning over long time horizons via temporal abstractions (Nachum et al., 2019; LeCun, 2022; Hafner et al., 2022; Gumbsch et al., 2023; Chen et al., 2024). Our method, `Puppeteer`, is also hierarchical in nature, but does not rely on skill primitives nor temporal abstraction for task learning. Instead, we learn a *single* low-level world model that can be reused across a variety of downstream tasks, and instead introduce a hierarchy in terms of data sources and input modalities.

# 6 CONCLUSION

We demonstrate that `Puppeteer` consistently produces motions that are considered natural and human-like by human evaluators compared to existing methods for visual RL, in an entirely data-driven manner and without any bells and whistles. These results are, to the best of our knowledge, unprecedented in the area of whole-body humanoid control. However, we acknowledge that our contributions have several limitations that may not be obvious: *(i)* while our proposed task suite consists of challenging visual whole-body control tasks with a detailed humanoid model, tasks primarily evaluate the visio-locomotive capabilities of current methods. We expect development of new tasks will be increasingly important as algorithms continue to improve, and we hope that the release of our benchmark will help facilitate that. *(ii)* Our hierarchical approach currently consists of two levels, only one of which is pretrained in the majority of our experiments. Our experiment on high-level pretraining suggests that it can be beneficial to pretrain both levels, but further research on how to pretrain and transfer the full hierarchical world model is warranted.

**Ethics statement.** All 51 participants in our user study are sourced from undergraduate and graduate student populations across multiple universities and disciplines on a volunteer basis. We do not collect personal or otherwise identifiable information about participants, and all participants have provided written consent to use of their responses for the purposes of this study. See Appendix F for more information.

**Reproducibility statement.** Code for method and environments, as well as model checkpoints, is made available at https://www.nicklashansen.com/rlpuppeteer. We rely on DM-Control and MuJoCo for simulation which are publicly available and licensed under the Apache 2.0 license. We leverage the MoCapAct dataset for pretraining which is also publicly available and licensed under the MIT license. Implementation details and a full list of hyperparameters are provided in Appendix D.

**Acknowledgements.** This work was supported, in part, by NSF CCF-2112665 (TILOS). Nicklas Hansen is supported by NVIDIA Graduate Fellowship, and Vlad Sobal is supported by NSF Award 19922658.

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

# A    ADDITIONAL QUALITATIVE RESULTS

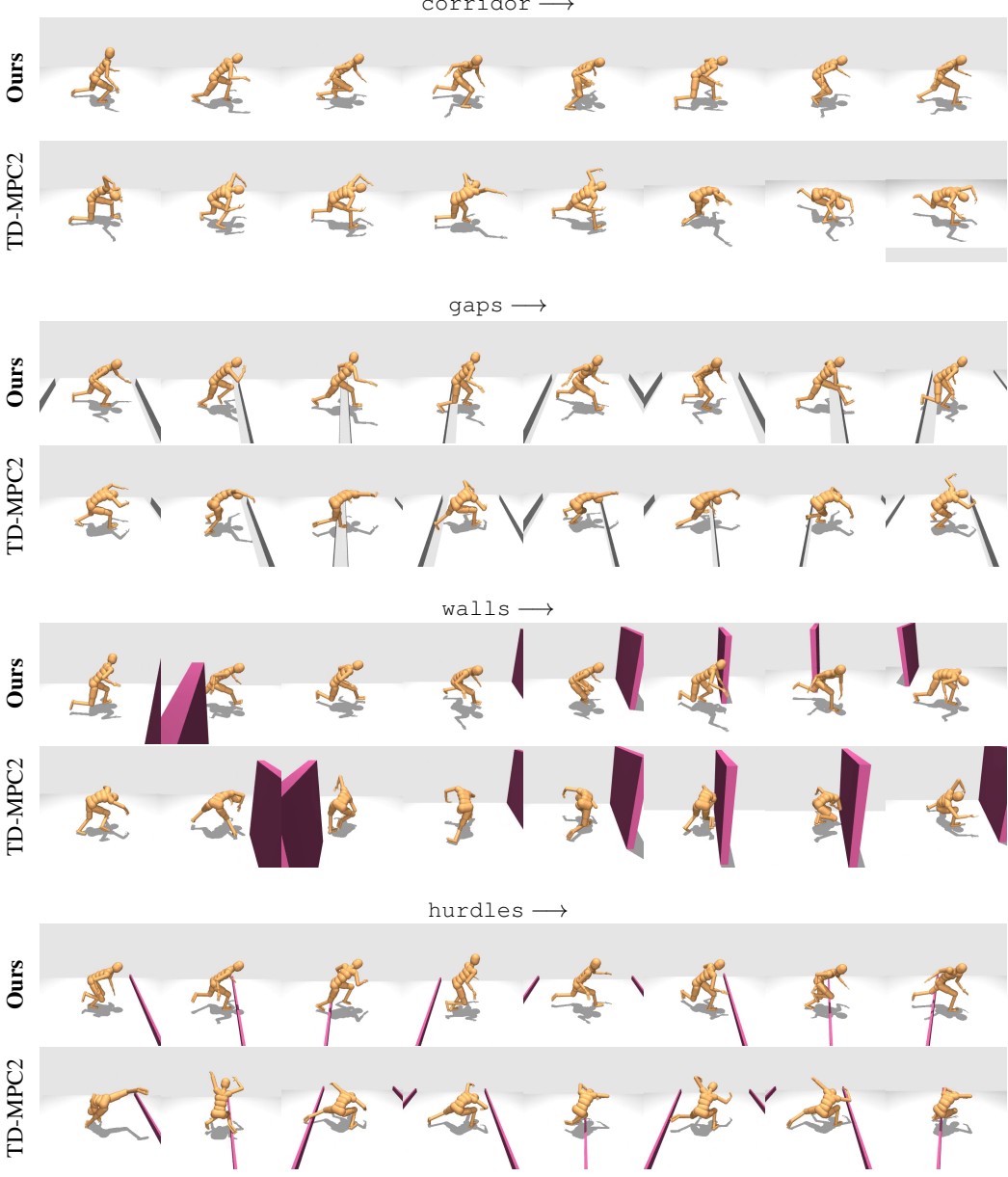

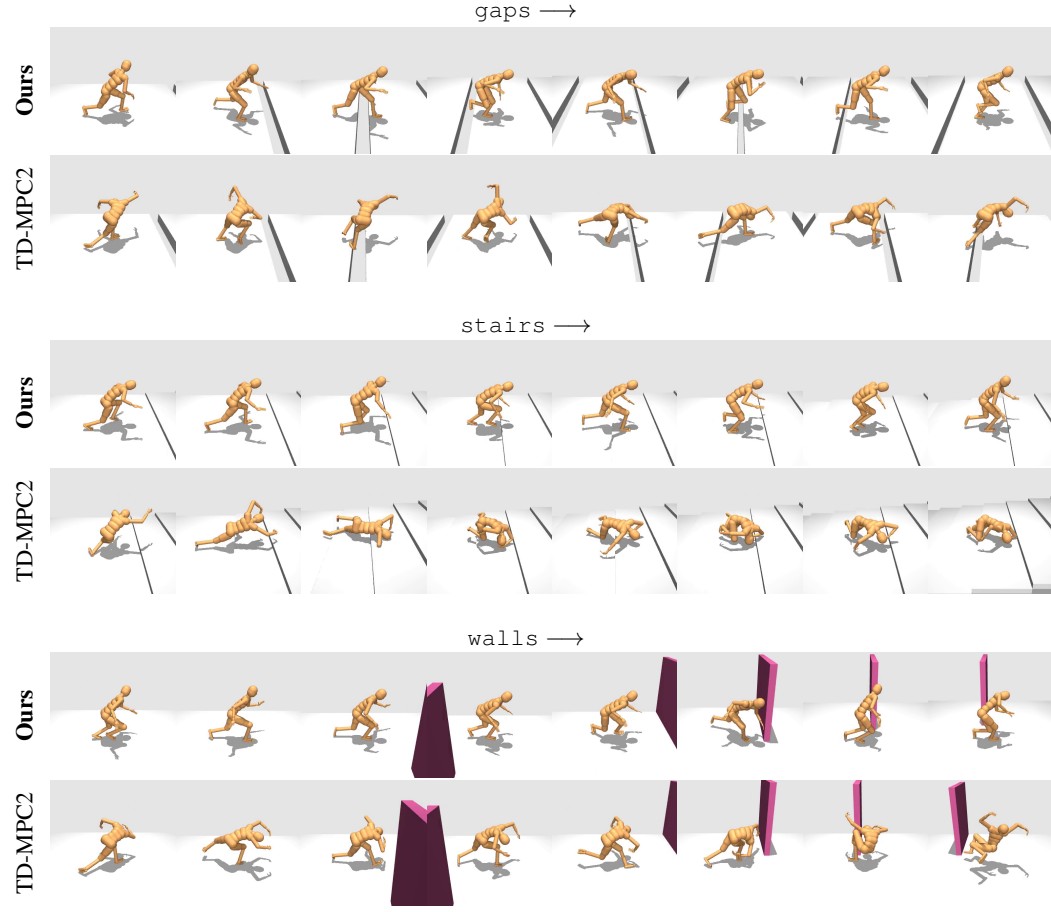

## B  ADDITIONAL QUANTITATIVE RESULTS

This section supplements our quantitative evaluations included in the main paper. In particular, Table 2 provides additional performance metrics for our low-level tracking agent, which we define as follows:

**Success rate** is defined as the percentage of clips that the tracking policy is able to track without ever exceeding a tracking error threshold of $0.5$ at any given time step. Because clips vary greatly in duration, we only evaluate success rate across at most 100 steps per clip by truncating longer clips.

**Tracking error** is defined as the mean distance between all joints and bodies in the humanoid wrt. the corresponding joints and bodies in the reference clip, evaluated on a per-time-step basis. Our definition of tracking error is consistent with that of Hasenclever et al. (2020) and Tassa et al. (2018).

**CoMic score** is the mean episode return achieved by the tracking policy as defined by the reward function proposed in CoMic (Hasenclever et al., 2020). We compute the CoMic score over all clips in the dataset, truncating clips to a maximum of 100 steps to prevent long clips from skewing results.

## C  SYSTEM REQUIREMENTS

Detailed system requirements for our method and baselines are included in Table 3.

*Table 2.* **Low-level tracking quality.** We report three additional metrics of tracking quality for the low-level tracking agent: success rate as defined by Luo et al. (2023), average tracking error across all 836 clips, and CoMic (Hasenclever et al., 2020) reward across all clips.

| | Success rate (%) ↑ | Tracking error ↓ | CoMic score ↑ |
|---|---|---|---|
| Offline only | 6.2 | 0.503 | 42.6 |
| 5% data | 74.4 | 0.260 | 45.4 |
| 25% data | 79.6 | 0.225 | 46.3 |
| 75% data | 87.9 | 0.202 | 47.4 |
| Ours | **88.3** | **0.187** | **48.7** |

*Table 3.* **System requirements.** Training wall-time, inference time, and GPU memory requirements for Puppeteer, TD-MPC2, and SAC on a single NVIDIA GeForce RTX 3090 GPU. Overall, wall-time and system requirements of Puppeteer are mostly comparable to that of TD-MPC2 across both state-based and visual RL. SAC runs approx. 3.6x faster than Puppeteer, but does not achieve any meaningful performance. We exclude replay buffer memory requirements for clarity, but note that 1M transitions require 14.1 GB memory for visual RL and 1.0 GB memory for state-based RL. We store the replay buffer in GPU memory such that CPU and RAM usage is negligible.

| | Wall-time (h / 1M steps) | Inference time (ms / step) | GPU memory (GB) |
|---|---|---|---|
| Puppeteer | 21.8 (vision: 29.0) | 88.2 | 0.6 |
| TD-MPC2 | 18.6 (vision: 25.2) | 50.8 | 0.5 |
| DreamerV3 | 50.2 | 18.7 | 3.9 |
| SAC | 5.9 | 2.2 | 0.4 |

# D  IMPLEMENTATION DETAILS

**MoCap dataset.** We use the "small" offline dataset provided by MoCapAct (Wagener et al., 2022), which is available at `https://microsoft.github.io/MoCapAct`. This dataset contains 20 noisy expert rollouts from each of 836 expert policies trained to track individual MoCap clips, totalling (suboptimal) 16,720 trajectories. Trajectories are variable length and are labelled with the CoMiC (Hasenclever et al., 2020) tracking reward which we use throughout this work. We solely use this dataset during (pre)training of the low-level tracking agent; the high-level puppeteering agent is trained independently of the tracking agent using only online interaction data and task rewards.

**Puppeteer.** We base our implementation off of TD-MPC2 and use default design choices and hyperparameters whenever possible. We experimented with alternative hyperparameters but did not observe any benefit in doing so. All hyperparameters are listed in Table 5. Our approach introduces only two new hyperparameters compared to prior work: loss coefficient for termination prediction (because our task suite has termination conditions; we add this to the TD-MPC2 baseline as well), and the number of low-level steps to take per high-level step (temporal abstraction).

**TD-MPC2.** We use the official implementation available at `https://github.com/nicklashansen/tdmpc2`, but modify the implementation to support multi-modal observations and termination conditions as discussed in Section 3. We empirically observe that TD-MPC2 degenerates to highly unrealistic behavior without a contact-based termination condition.

**SAC.** We benchmark against the implementation from `https://github.com/denisyarats/pytorch_sac` (Yarats & Kostrikov, 2020) due to its strong performance on lower-dimensional DMControl tasks as well as its popularity among the community. We modify the implementation to support early termination. We experiment with a variety of design choices and hyperparameters as we find vanilla SAC to suffer from numerical instabilities on our task suite (presumably due to high-dimensional observation and action spaces), but are unable to achieve non-trivial performance. The ablation in Figure 8 (hierarchical planning) strongly suggests that planning is a key driver of performance in Puppeteer and TD-MPC2, while SAC is a model-free method incapable of planning. Design choices and hyperparameters that we experimented with are as follows:

*Table 4.* **List of SAC design choices and hyperparameters.** We experiment with a variety of design choices and hyperparameters, but find that they all fail to achieve non-trivial performance.

| Design choice | Values |
|---|---|
| Number of $Q$-functions | $2, 5$ |
| TD-target | Default, REDQ (Chen et al., 2021) |
| Activation | ReLU, Mish, LayerNorm + Mish |
| MLP dim | $256, 512, 1024$ |
| Batch size | $256, 512$ |
| Learning rate | $3 \times 10^{-4}, 1 \times 10^{-3}$ |

**DreamerV3.** We use the official implementation available at `https://github.com/danijar/dreamerv3`, and use the default hyperparameters recommended for proprioceptive DMControl tasks. A key selling point of DreamerV3 is its robustness to hyperparameters across tasks (relative to SAC), but we find that DreamerV3 does not achieve any non-trivial performance on our task suite. While DreamerV3 is a model-based algorithm, it does not use planning, which the ablation in Figure 8 (hierarchical planning) finds to be a key driver of performance in Puppeteer and TD-MPC2.

*Table 5.* **List of hyperparameters.** We use the same hyperparameters across all tasks, levels (high-level and low-level), and across both Puppeteer and TD-MPC2 when applicable. Hyperparameters unique to Puppeteer are  highlighted .

| Hyperparameter | Value |
|---|---|
| **Planning** | |
| Horizon ($H$) | 3 |
| Iterations | 8 |
| Population size | 512 |
| Policy prior samples | 24 |
| Number of elites | 64 |
| Temperature | 0.5 |
| Low-level steps per high-level step | 1 |
| | |
| **Policy prior** | |
| Log std. min. | $-10$ |
| Log std. max. | 2 |
| | |
| **Replay buffer** | |
| Capacity | $1,000,000$ |
| Sampling | Uniform |
| | |
| **Architecture** | |
| Encoder dim | 256 |
| MLP dim | 512 |
| Latent state dim | 512 |
| Activation | LayerNorm + Mish |
| Number of $Q$-functions | 5 |
| | |
| **Optimization** | |
| Update-to-data ratio | 1 |
| Batch size | 256 |
| Joint-embedding coef. | 20 |
| Reward prediction coef. | 0.1 |
| Value prediction coef. | 0.1 |
| Termination prediction coef. | 0.1 |
| Temporal coef. ($\lambda$) | 0.5 |
| $Q$-fn. momentum coef. | 0.99 |
| Policy prior entropy coef. | $1 \times 10^{-4}$ |
| Policy prior loss norm. | Moving $(5\%, 95\%)$ percentiles |
| Optimizer | Adam |
| Learning rate | $3 \times 10^{-4}$ |
| Encoder learning rate | $1 \times 10^{-4}$ |
| Gradient clip norm | 20 |
| Discount factor | 0.97 |
| Seed steps | 2,500 |

# E  TASK SUITE

We propose a benchmark for visual whole-body humanoid control based on the "CMU Humanoid" model from DMControl (Tassa et al., 2018). Our simulated humanoid has 56 fully controllable joints ($\mathcal{A} \in \mathbb{R}^{56}$), and includes both head, hands, and feet. Actions are normalized to be in $[-1, 1]$. Our task suite consists of 5 vision-conditioned whole-body locomotion tasks (corridor, hurdles, walls, gaps, stairs), as well as 3 tasks that use proprioceptive information only (stand, walk, run). All 8 tasks are illustrated in Figure 4.

Observations always include proprioceptive information, as well as either visual inputs (high-level agent) or a command (low-level agent). The proprioceptive state vector is 212-dimensional and consists of relative joint positions and velocities, body velocimeter and accelerometer, gyro, joint torques, binary touch (contact) sensors, and orientation relative to world $z$-axis. Visual inputs are raw $64 \times 64$ RGB images captured by a third-person camera (as seen in Figure 4) without any preprocessing steps, and tracking commands are 15-dimensional vectors (corresponding to 5 points in 3D space) with values in $[-1, 1]$.

Downstream task reward functions are based on the humanoid reward functions in DMControl with minimal modification to fit our higher DoF embodiment. All 5 visual tasks use the same reward function, which is proportional to forward velocity of the humanoid and is bounded to always be non-negative:

$$R(\mathbf{s}) \doteq \text{clip}(\text{linvel}_x, [0, v_{\text{target}}]) \tag{3}$$

where $\text{linvel}_x$ is linear velocity along the $x$-axis, and the clip operator bounds the reward value to always be non-negative and at most that of a target velocity $v_{\text{target}}$ which we set to 6 in all tasks. The 3 proprioceptive tasks use a similar reward function, except that the agent is rewarded for velocity in any $XY$-direction, and has an additional term that encourages an upright pose:

$$R(\mathbf{s}) \doteq \min(|\text{linvel}_{xy}|, v_{\text{target}}) + \alpha \cdot \text{headpos}_z \tag{4}$$

where $\alpha$ is a constant coefficient balancing the two reward terms, and $\text{headpos}_z$ is the height of the humanoid head in the world frame. The additional height reward term is adopted from the stand, walk, and run run tasks that DMControl implement with a simplified humanoid model ($\mathcal{A} \in \mathbb{R}^{24}$). We find that the TD-MPC2 baseline produces very unrealistic behaviors without the additional reward term, so we choose to keep the term to make comparison more fair.

# F  USER STUDY

To compare the "naturalness" of policies learned by our method vs. TD-MPC2, we design a user study in which humans are asked to watch short ($\sim$10s) pairs of clips of simulated humanoid motions generated for each of our 5 visual whole-body humanoid control tasks. Each user is presented with 2 such pairs per task, totalling 10 pairs per user. Sample clips used in the user study are available at https://www.nicklashansen.com/rlpuppeteer, as well as in Appendix A. Pairs are generated by converged Puppeteer and TD-MPC2 agents. We generate 5 rollouts per task for each of two separately trained agents (random seed 1 and 2) using the same method (*i.e.*, Puppeteer or TD-MPC2), and select the clips with median episode return for each of the two random seeds. We use clips generated by two unique random seeds to ensure that diversity in behavior due to inter-seed variability is captured in the user study, and we select the median clip to ensure that we neither favor nor disadvantage a method due to outliers. The concrete instructions provided to users in the study are as follows:

**Instructions**

In this study, you will watch pairs of short ($\sim$10 seconds) clips of simulated humanoid motions. For each pair, you are asked to determine which of the two clips appear more "natural" and "human-like" to you, i.e., which clip looks more like the behavior of a real human.

Users are then provided with each of the 10 pairs of clips, and prompted to answer questions of the form:

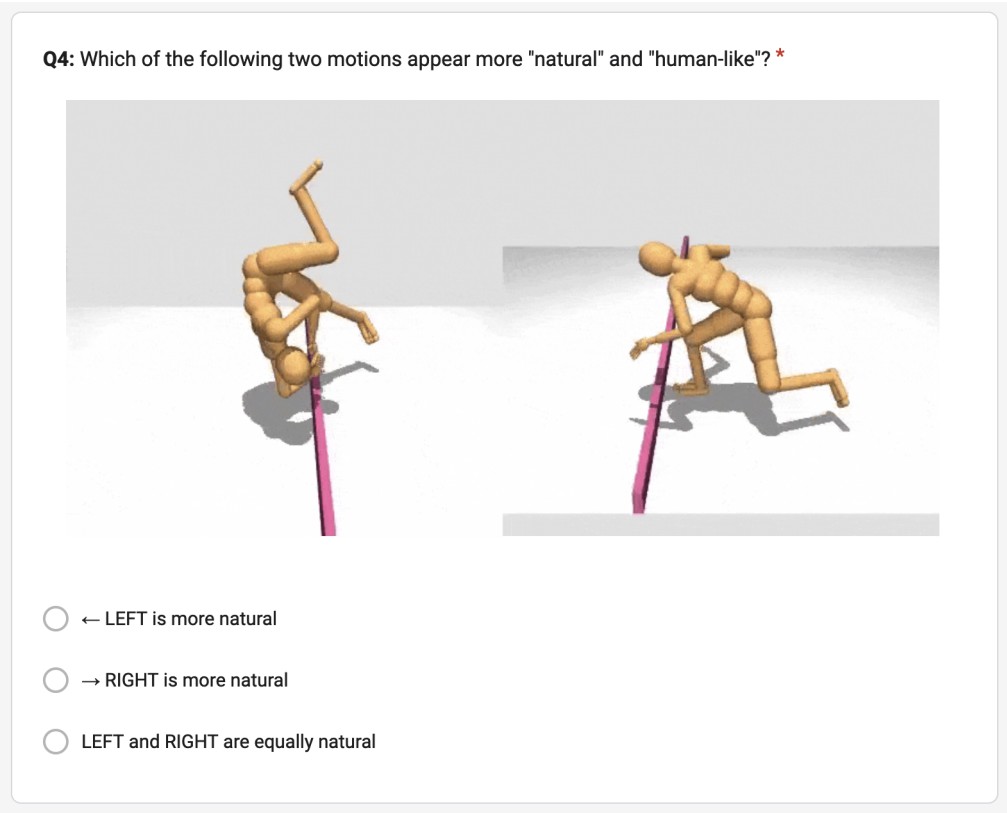

*Figure 10.* **Screenshot of a question from the user study.** Users are shown two clips side-by-side and are asked to provide their preference.

**Q1:** Which of the following two motions appear more "natural" and "human-like"?

1. ← LEFT is more natural
2. → RIGHT is more natural
3. LEFT and RIGHT are equally natural

The order of clips is selected at random for each pair. Aggregate results from the user study are provided in Table 6, and Figure 10 shows a sample question from the user study. Participants are sourced from undergraduate and graduate student populations across multiple universities and disciplines on a volunteer basis. We do not collect personal or otherwise identifiable information about participants, and all participants have provided written consent to use of their responses for the purposes of this study.

*Table 6.* **Results from the user study.** We summarize results from our user study ($n = 51$) below by reporting per-pair aggregate numbers. Higher is better ↑. Clips generated by our method, `Puppeteer`, are considered more natural by a super-majority of participants.

| Pair | TD-MPC2 | Equal | Ours |
|---|---|---|---|
| **Corridor** | | | |
| Pair 1 | 0 | 0 | 51 |
| Pair 2 | 0 | 2 | 49 |
| | | | |
| **Hurdles** | | | |
| Pair 1 | 0 | 0 | 51 |
| Pair 2 | 0 | 0 | 51 |
| | | | |
| **Walls** | | | |
| Pair 1 | 1 | 2 | 48 |
| Pair 2 | 0 | 0 | 51 |
| | | | |
| **Gaps** | | | |
| Pair 1 | 0 | 0 | 51 |
| Pair 2 | 0 | 0 | 51 |
| | | | |
| **Stairs** | | | |
| Pair 1 | 0 | 2 | 49 |
| Pair 2 | 1 | 3 | 47 |
| | | | |
| **Aggregate** | 0.4% | 1.8% | 97.8% |

