# OpenReview forum: "Hierarchical World Models as Visual Whole-Body Humanoid Controllers"
_ICLR.cc/2025/Conference — ICLR 2025 Poster_

### Official Review · Reviewer_EisB · 2024-11-02

**Soundness:** 3
**Presentation:** 3
**Contribution:** 2
**Rating:** 5
**Confidence:** 4

**Summary:**

The paper presents a novel hierarchical world model Puppeteer designed for visual whole-body humanoid control, which operates without relying on predefined skill primitives, reward designs, or temporal abstractions. This paper also introduces a new task suite consisting of eight challenging tasks for evaluating humanoid control, demonstrating that Puppeteer produces more natural and human-like motions preferred by human evaluators in comparison with model-free and model-based RL baselines.

**Strengths:**

* This paper is well-organized, well-written, and includes clear figures.
* A well-designed hierarchical control framework (although the idea is not novel) is implemented to control humanoid motion in a more natural way: the high-level agent generates commands given visual observations, and the low-level agent is responsible for executing them.
* The proposed visual whole-body high-dimensional humanoid control benchmark enrich the evaluation platforms in the area.

**Weaknesses:**

* The paper primarily evaluates the visio-locomotive capabilities of the humanoid model. It could be better to expand the range of tasks to include more diverse scenarios that test different aspects of humanoid capabilities.
* More baseline method should be compared, like HumanoidOlympics (https://arxiv.org/pdf/2407.00187). This paper also uses human motion data and reinforcement learning to train natural humanoid motions in various tasks.

**Questions:**

* Although the motion produced by Puppeteer is more natural, why does the humanoid robot always lean forward while moving?
* Can the proposed method be validated in real-world experiments?
* Can this framework be generalized to manipulation tasks in HumanoidOlympics?

---

> ### Author Response · Authors · 2024-11-22
> **Thank you!**
>
> We thank the reviewer for their valuable feedback. We address your comments in the following.
>
> ----
>
> **Q:** Although the motion produced by Puppeteer is more natural, why does the humanoid robot always lean forward while moving?
>
> **A:** It is difficult to give a definitive answer to this question, but we conjecture that it may be one of two reasons, or more likely a combination of the two: (1) leaning forward with the torso creates momentum while being a relatively stable gait so it is possible that the high-level policy learns such behavior for this reason, and (2) there are many instances in the human MoCap dataset where the human leans forward while performing highly dynamic motions, which could bias the low-level tracking policy towards such a gait. For context, we include videos of various reference motions from the MoCap dataset along with our learned tracking policy on our project webpage; you will notice that the human reference motion in clip 9 (bottom row, second to right) leans forward while making a quick turn. We agree that it would be very interesting to study how different mixtures of MoCap data (e.g. excluding all “running” clips from the pretraining data) influence the behavior of learned policies, and we will be happy to include some exploratory results on this in the camera-ready version if the reviewer believes it would add value to the manuscript.
>
> ----
>
> **Q:** Can the proposed method be validated in real-world experiments?
>
> **A:** We agree that deploying our method on a real humanoid robot is a natural next step (which we intend to pursue), but believe that it would be outside of the scope of this manuscript. Deploying a learned policy on a relatively new hardware platform such as the Unitree H1 / G1 will require substantial hardware engineering and sim-to-real design effort, while our work focuses more on the algorithmic foundations. That said, we do believe deploying our method to real hardware would be possible without significant algorithmic changes.
>
> ----
>
> **Q:** Can this framework be generalized to manipulation tasks in HumanoidOlympics?
>
> **A:** Yes, this should indeed be possible, and we are interested in pursuing this in the future. However, we would like emphasize two things: (1) HumanoidOlympics is a strictly state-based benchmark whereas our method and benchmark is designed specifically for visuo-motor control (image inputs), and (2) both HumanoidOlympics and HumanoidBench (the two alternatives to our benchmark) were developed concurrently with our work.
>
> ----
>
> We believe that our response addresses your main concerns. However, if that is not the case please let us know if you have any additional comments. We would greatly appreciate it if the reviewer could provide us with precise and actionable feedback such that we can fully address your concerns. Thanks again for your time and valuable feedback!

---

> > ### Author Response · Authors · 2024-11-24
> > **A gentle reminder**
> >
> > Dear reviewer EisB,
> >
> > The end of the discussion period is rapidly approaching, and we would really appreciate it if you had a moment to look over our response and changes to the manuscript. We believe that your main concerns are addressed by our response, but if not we will be more than happy to work with you on further addressing them. Thanks again for your time and for serving as a reviewer!

---

> ### Comment · Reviewer_EisB · 2024-11-25
>
> Thanks for your reply! However, as the authors do not conduct any experiment I proposed in the original review, I change my scores as follows: Rating: 5 and Confidence: 4. I believe HumanoidBench (https://arxiv.org/abs/2403.10506) is not a concurrent work, and this paper should include manipulation tasks, as they would be much more suitable and easier to evaluate motion naturalness than the locomotion tasks designed in the paper.
>
> Furthermore, I do not think the results of locomotion naturalness are compelling enough to support the major claims in the paper especially the authors do not really compare the motion clips produced by this method with the real motion capture data in the user study but only compare with regular RL method.

---

> > ### Author Response · Authors · 2024-11-26
> > **Thank you for your response**
> >
> > Thank you for your response! We really appreciate your feedback and engagement during this discussion period. We respond to each of your comments in the following.
> >
> > > I believe HumanoidBench (https://arxiv.org/abs/2403.10506) is not a concurrent work
> >
> > HumanoidBench is still an unpublished work and only a preprint was made available on arXiv (concurrent with the development of our benchmark and method). Regardless, while we agree that HumanoidBench is an interesting resource for robot learning research, it does not consider visual observations (which is a key focus of our approach) and the Unitree H1 robot used in HumanoidBench is significantly lower DoF than the humanoid model that we consider in this work. The same is true for the HumanoidOlympics benchmark that was referenced in your original review. Additionally, the primary contribution of these papers is the simulation environments that they propose, whereas our paper contributes both a set of visuo-motor control tasks *and* a new RL algorithm for visual whole-body humanoid control.
> >
> > > and this paper should include manipulation tasks, as they would be much more suitable and easier to evaluate motion naturalness than the locomotion tasks designed in the paper.
> >
> > Respectfully, does the reviewer have any scientific evidence that human-like locomotion is a less important problem than human-like manipulation? We ask the reviewer to please judge our work based on scientific merit rather than personal preference or feelings.
> >
> > > the authors do not really compare the motion clips produced by this method with the real motion capture data
> >
> > While it is true that we do not compare to the real MoCap clips in our user study, this is primarily because the focus of our study is natural motion in *downstream tasks* rather than imitation of human clips. There are no suitable MoCap clips available for the types of tasks that we consider in this work, hence their exclusion from our user study. That said, we do compare our method against real MoCap clips in the pretraining stage, both qualitatively (clips are available on our project page under "tracking results") as well as quantitatively (success rate, average tracking error, and CoMic score). We observe that e.g. availability of more pretraining data results in more accurate tracking of reference motions from real humans.

---

> > > ### Comment · Reviewer_EisB · 2024-11-26
> > >
> > > Thanks for your reply!
> > >
> > > As the paper is designed for "whole-body humanoid control", I still believe it is important to showcase the proposed method's performance in manipulation task. Could you design one simpliest manipulation task and demonstrate the motion generated by the proposed method has better qualities and more natural movements than traditional RL baselines? I will raise the score to acceptance if the results are provided.

---

> > > > ### Author Response · Authors · 2024-12-01
> > > >
> > > > Thank you for the continued discussion, we really appreciate it. To address your comments:
> > > >
> > > > > I still believe it is important to showcase the proposed method's performance in manipulation task
> > > >
> > > > We agree that extending our work to manipulation tasks would be really interesting and a natural next step in whole-body humanoid control research. However, we would like to point out that the human mocap dataset (CMU Motion Capture Database [1], and by extension MoCapAct [2] which retargets the mocap data to our specific humanoid model) that we rely on in this work does not contain any object manipulation. While that does not preclude the possibility of pretraining our method on this dataset and transferring it to downstream tasks that involve object manipulation, the pretraining data contains no human prior of how to interact with objects (e.g. grasping). We do not believe that this is a limitation of our method itself, but rather **a limitation of the type of human mocap data that is currently available** to the public.
> > > >
> > > > > As the paper is designed for "whole-body humanoid control"
> > > >
> > > > We do believe this to be an accurate claim regardless of whether our benchmark tasks include object manipulation or not. We have demonstrated that our method is able to control a full 56-DoF humanoid model from visual inputs, and produces natural and human-like motions when pretrained on human mocap data. It is evident from our qualitative results (videos are available on our [project webpage](https://rlpuppeteer.github.io)) that our method is capable of controlling *the whole body* and accurately tracks diverse reference motions and poses. While it is unfortunate that no object-centric mocap dataset exists yet for whole-body humanoid control, there is no technical reason for why our method could not be extended to such a setting if it became available.
> > > >
> > > > In summary, we recognize the validity of the reviewer's suggestion yet urge the reviewer to please judge our work based on its technical contributions and scientific merit as a whole. Thanks again for your time and valuable feedback!
> > > >
> > > > ----
> > > >
> > > > [1] Carnegie Mellon University Graphics Lab Motion Capture Database, URL: *http://mocap.cs.cmu.edu* (2003)
> > > >
> > > > [2] Wagener, N., Kolobov, A., Frujeri, F. V., Loynd, R., Cheng, C., Hausknecht, M., "MoCapAct: A Multi-Task Dataset for Simulated Humanoid Control", NeurIPS 35 (2022)

---

> > > > > ### Author Response · Authors · 2024-12-02
> > > > > **Discussion ends TODAY**
> > > > >
> > > > > Dear reviewer,
> > > > >
> > > > > This is a friendly reminder that the discussion period is ending **TODAY**, December 2. We hope that our previous response addresses your concern regarding manipulation tasks to the extent possible, and we would really appreciate it if you could take a moment today to reevaluate our contributions and revised manuscript based on our discussion as well as all the changes that we have made per other reviewers' request. Thanks again for your time and valuable feedback!
> > > > >
> > > > > Best,
> > > > >
> > > > > Authors of Puppeteer

---

> > > > > > ### Comment · Area_Chair_kWAX · 2024-12-02
> > > > > >
> > > > > > Dear Reviewer EisB,
> > > > > >
> > > > > > As the discussion period ends today, please be sure to read and reply to the authors' response to your request for results involving a manipulation task.
> > > > > >
> > > > > > Best,\
> > > > > > AC

---

> > > > > > > ### Author Response · Authors · 2024-12-03
> > > > > > > **Final reminder**
> > > > > > >
> > > > > > > Dear reviewer EisB,
> > > > > > >
> > > > > > > This is a **final reminder** that the discussion period ends in approx. **6 hours** (Dec 2, anywhere on earth). We would really appreciate it if the reviewer would take a moment to read our previous response and let us know whether it addresses the reviewer's concerns. We have worked hard to accommodate the requested changes of all reviewers, and we hope that the reviewer would consider revising their score as a result.
> > > > > > >
> > > > > > > Best,
> > > > > > >
> > > > > > > Authors of Puppeteer

---

### Official Review · Reviewer_2bf6 · 2024-11-03

**Soundness:** 3
**Presentation:** 3
**Contribution:** 2
**Rating:** 5
**Confidence:** 5

**Summary:**

Post rebuttal | the authors did a good job in the rebuttal phase to include some necessary baselines.

The paper improved a lot during the rebuttal phase and now includes some of the necessary baselines to validate the method. When I asked the authors about measures to ensure fairness in the new baseline experimentation, they argued that the considered MBRL algorithms are known to be robust to hyperparameter changes, which is true but not a valid argument for a simple reason: if an algo is robust to hyperparameter choice, doesn't mean that it cannot benefit from tuning. As I believe the proposed method was subject to some tuning during development time (like any ML method is), measures to ensure fairness of comparison would be necessary. Additionally, the baselines added during rebuttal just scratch the surface of hierarchical RL approaches and are themselves made up baselines combining known high-level algorithms with low-level TD-MPC. While these baselines are very important and necessary to understand the method, including standard baselines from the literature would strongly improve the paper and increase its impact. I highly encourage the authors to do that. I will be raising my score but only to a 5, as I still think the paper is not ready for publication at ICLR, despite the good improvements.

---

The paper proposes "Puppeteer" a hierarchical decision-making approach tailored for visual whole-body humanoid control. The proposed method trains two separate world models for high-level and low-level control purposes. The low-level world model is concerned with tracking reference joint-level trajectories produced by the high-level controller. The high-level controller can additionally be conditioned on visual data. Both world models are based on TD-MPC2 which is a sampling-based MPC approach with learned decoder-free world models (with deterministic-only components) and a learned value function for long-horizon value assignment. TD-MPC2 is further extended to include a termination encoder head as is common in other model-based RL methods such as dreamer. The paper claims that the proposed method achieves results that are mostly comparable to TD-MPC2's results, while the plots show significantly worse results than TD-MPC2 in terms of asymptotic performance. The main advantage of the method is that it produces more natural and human-like motion, which was quite well shown in the experiments. The paper also ablates multiple design choices.

**Strengths:**

- the paper is generally well-written and an enjoyable read, I also liked the figures and plots.
- the proposed approach is very interesting and promising and is a natural next step to extend the TD-MPC2 framework.
- the method is evaluated on multiple humanoid tasks including environments with only proprioception as well as others with additional visual observations.
- the ablations nicely evaluate the role of the different design choices of the method. I especially appreciate the study of the role of planning in the architecture.
- the baselines include model-free and model-based approaches.

**Weaknesses:**

- the method section misses a detailed motivation for why hierarchy improves the naturalness of the motions.
- the method section misses a detailed explanation concerning the exact usage of the high-level commands (see question 2).
- the paper introduces a hierarchical version of td-mpc2, the baselines however do not include a single hierarchical RL approach, I would at least consider including a hierarchical implementation of dreamer [1].
- [main weakness] The results of the paper are weak, at least in the current way in which they are presented. While the method is interesting and makes sense, the results show that it significantly underperforms TD-MPC2 but improves the naturalness of the produced motions. That would have been an acceptable tradeoff if the paper could justify why the proposed method improves the naturalness of the motions with intuitions and ideally, some experiments that validate them.

** Minor issues:**

- line 079 end-effector joints --> end-effector links.
- punctuation is missing in the equations (but I understand that this is a matter of style, so no pressure).

Overall the paper proposes an interesting approach, but it currently fails to showcase the benefits of this approach. I am willing to raise my score if this aspect is properly addressed.

**Questions:**

- the main advantage of this method over TD-MPC2 is the resulting naturalness of the motions. Can the authors elaborate on why they think the proposed method improves this aspect? (here I mean further explaining the reward hacking argument made in the paper and perhaps including other arguments that could make sense)
- can the authors elaborate on how the low-level policies exactly track the high-level commands? Since the low-level receives a sequence of commands does it keep using $c_t$ until the tracking error is below a threshold, or is it only used for a single step independent of the outcome of applying the one-step action?
- on the methods side of things, the paper extends td-mpc2 to a hierarchical architecture. Can the authors compare the method to the hierarchical version of Dreamer [1]?

[1] Hafner, Danijar, et al. "Deep hierarchical planning from pixels." Advances in Neural Information Processing Systems 35 (2022)

---

> ### Author Response · Authors · 2024-11-22
> **Thank you!**
>
> We thank the reviewer for their valuable feedback. We address your comments in the following.
>
> ----
>
> **Q:** the main advantage of this method over TD-MPC2 is the resulting naturalness of the motions. Can the authors elaborate on why they think the proposed method improves this aspect?
>
> **A:** We attribute the naturalness of motions to pretraining on a large collection of human MoCap clips. By first training a single, reusable tracking (low-level) agent to track reference motions from this dataset, and then subsequently training a task-specific high-level agent to perform downstream tasks by providing commands to the low-level agent, our method implicitly leverages the behavioral priors obtained during pretraining. This is, to some extent, validated by our ablations in Figure 8 that vary the number of MoCap clips used during pretraining – we observe a strong correlation between number of MoCap clips and tracking performance. Lastly, per request of reviewer HShf we have added 4 new evaluations of our low-level tracking agent and its relevant ablations, which are included in our reply here [https://openreview.net/forum?id=7wuJMvK639&noteId=y6ra8KpCFC](https://openreview.net/forum?id=7wuJMvK639&noteId=y6ra8KpCFC) as well as in Appendix B of our updated manuscript.
>
> ----
>
> **Q:** can the authors elaborate on how the low-level policies exactly track the high-level commands? Since the low-level receives a sequence of commands does it keep using until the tracking error is below a threshold, or is it only used for a single step independent of the outcome of applying the one-step action?
>
> **A:** The low-level agent is provided a command (or sequence of commands) by the high-level policy at every step, and both levels will replan a sequence of high-level commands + low-level actions regardless of whether the previous command was achieved or not. This design decision is consistent with the pretraining phase in which the tracking agent is trained to track a reference motion. We have added the sentence “The high-level policy outputs commands at a fixed frequency regardless of whether the previous command was achieved” to Section 3.2 (L215-216) which can be found in the updated manuscript. We hope that this makes it more clear, and thank the reviewer for pointing it out.
>
> ----
>
> **Q:** on the methods side of things, the paper extends td-mpc2 to a hierarchical architecture. Can the authors compare the method to the hierarchical version of Dreamer [1]?
>
> **A:** We agree that this would be an interesting comparison in principle. However, our empirical results in Figure 5 indicate that DreamerV3 does not achieve any meaningful performance even on the state-based humanoid control tasks (same as SAC), while TD-MPC2 converges in <1M environment steps. These results are in line with concurrent work [2] that also finds TD-MPC2 to outperform DreamerV3 by a large margin in state-based humanoid control. We thus decided to proceed with TD-MPC2 as our base algorithm in the remainder of our experiments which are all vision-based. We will be happy to include results for a hierarchical DreamerV3 in the camera-ready version if the reviewer believes that this will be informative, but we do not expect results to change much compared to the non-hierarchical version.
>
> [2] Sferrazza, C., Huang, D., Lin, X., Lee, Y., Abbeel, P., “HumanoidBench: Simulated Humanoid Benchmark for Whole-Body Locomotion and Manipulation“ (2024)
>
> ----
>
> We believe that our response addresses your main concerns. However, if that is not the case please let us know if you have any additional comments. We would greatly appreciate it if the reviewer could provide us with precise and actionable feedback such that we can fully address your concerns. Thanks again for your time and valuable feedback!

---

> > ### Author Response · Authors · 2024-11-24
> > **A gentle reminder**
> >
> > Dear reviewer 2bf6,
> >
> > The end of the discussion period is rapidly approaching, and we would really appreciate it if you had a moment to look over our response and changes to the manuscript. We believe that your main concerns are addressed by our response, but if not we will be more than happy to work with you on further addressing them. Thanks again for your time and for serving as a reviewer!

---

> ### Comment · Reviewer_2bf6 · 2024-11-24
> **Reply to authors**
>
> Thank you for your rebuttal. Unfortunately, the current form of the paper and the answers from the rebuttal fail to convey the contribution of this work. The paper mainly advances the naturalness of the humanoid motion. The reason for the emergence of this more natural motion (according to the authors) is the pretraining of the low-level tracking agent on a human MoCap dataset, which represents expert interactions with a very narrow distribution. Without extensive comparison of the proposed approach to other hierarchical frameworks that should equally leverage the MoCap dataset, the value of this work is unclear. Model-based methods are typically advantageous due to their sample efficiency. However, it is unclear whether this advantage remains in the presence of a dataset that can facilitate pretraining a low-level tracking policy. Such low-level trackers reduce the complexity of the problem and the advantage of using a TD-MPC style high-level planner is unclear. I advise the authors to include a comparison with other hierarchical frameworks based on Dreamer, as well as model-free methods. Otherwise, the value of this work is unclear. The work looks at an interesting problem and could potentially make a nice contribution. However, in its current form, it is not ready for publication and I still propose rejecting it unless the mentioned baselines are included in the comparison.

---

> > ### Author Response · Authors · 2024-11-25
> > **We appreciate your feedback**
> >
> > Thank you for the constructive feedback, we really do appreciate your time and effort! It appears that your primary concern at this point is our choice of baselines. We would like to take a moment to clarify why we picked the current baselines and ablations, as well as provide new empirical results along the lines of what you are suggesting.
> >
> > **Choice of baselines:** Our goal is to learn visual whole-body humanoid controllers that perform well (data-efficient and strong asymptotic performance) *and* produce natural motions. We first benchmarked three common algorithms for continuous control: SAC, DreamerV3, TD-MPC2, and found that only TD-MPC2 achieved non-trivial performance. Based on our ablations in Figure 8 it appears that the planning component of TD-MPC2 is critical to performance on these high-dimensional tasks, which could explain the poor performance of SAC and DreamerV3 as neither of them use planning. We thus decided to use TD-MPC2 as the backbone learning algorithm at both levels of our hierarchical approach. Regarding your specific comment that
> >
> > > the advantage of using a TD-MPC style high-level planner is unclear
> >
> > we would like to again reference our planning ablation in Figure 8; this result clearly demonstrates that performance of our method degrades substantially if planning is disabled at either level (i.e., planning at the high-level is necessary). This brings us to our second point.
> >
> > **New empirical results:** To further confirm the importance of a TD-MPC style high-level planner, we run a new set of experiments that use the **same** low-level tracking agent as our approach (based on TD-MPC2) but replaces the high-level TD-MPC2 agent with a SAC policy. A revised Figure 5 (learning curves) can be found [on this link](https://i.imgur.com/tcxIyqT.png); we denote this ablation as "SAC w/ LL. TD-MPC2" and run a full set of experiments (10 seeds) on the three state-based tasks *stand*, *walk*, and *run*. Similar to the non-hierarchical SAC baseline and our non-planning ablation, the high-level SAC agent fails to achieve any meaningful performance on these three tasks, and often suffers from numerical instability (divergence). In light of these results, we do not believe that a baseline with SAC at both levels will provide any additional insights. We will be very happy to include the equivalent "DreamerV3 w/ LL. TD-MPC2" ablation as well, but this will take a little while longer since DreamerV3 runs significantly slower than SAC.
> >
> > We have updated our manuscript to include these new results, as well as a description of the added baseline on L315-317. We hope that our response helps address your concern regarding baselines! Either way, thanks again for your time and valuable feedback.

---

> > > ### Comment · Reviewer_2bf6 · 2024-11-25
> > > **Reviewer reply**
> > >
> > > Thank you for the quick follow-up and clarifications. This baseline is a step in the right direction but is not sufficient on its own as SAC typically requires a substantially higher amount of samples than MBRL methods. Can you please include a dreamer-based baseline? A smaller number of seeds would be acceptable for the short rebuttal period.

---

> ### Author Response · Authors · 2024-11-25
>
> We are pleased to hear that our ablation with TD-MPC2 at the low level and SAC at the high level addresses your concern to some extent.
>
> We are currently running the same ablation with DreamerV3 at the high level, but these experiments (3 tasks, 10 seeds each) will take a while since DreamerV3 is *significantly* slower than SAC/TD-MPC2. We estimate 3M environment steps (as is reported for our other methods) to take **approx. 6 days**. We will keep you posted with results as training progresses.
>
> In the mean time, we would like to provide some additional context and push back on this claim a bit:
>
> > This baseline is a step in the right direction but is not sufficient on its own as SAC typically requires a substantially higher amount of samples than MBRL methods.
>
> We do not believe there is substantial evidence for this in existing literature in the context of SAC vs. DreamerV3. The DreamerV3 paper does **not** compare against SAC at all, and does **not** consider humanoid control tasks, but several other related works have made this comparison. In particular, the TD-MPC2 paper benchmarks SAC and Dreamer-V3 on DMControl (including humanoids albeit with a lower DoF than we consider in this work) and find that DreamerV3 and SAC mostly have comparable data-efficiency but that DreamerV3 tends to converge to a higher asymptotic performance: approx. 750 vs. 600 mean reward on DMControl excluding Humanoid tasks, and approx. 850 vs. 500 reward on the Humanoid Walk task of DMControl. For reference, TD-MPC2 achieves approx. 900 reward on Humanoid Walk in 2M environment steps, whereas DreamerV3 achieves its 850 reward at around 12M environment steps. These numbers are echoed by Humanoid-Bench, which likewise benchmark SAC, DreamerV3, TD-MPC2 on state-based humanoid locomotion tasks. On this benchmark, TD-MPC2 performs significantly better than either method both in terms of asymptotic performance and data-efficiency, and SAC + DreamerV3 generally have similar convergence rate but DreamerV3 has higher asymptotic performance across the board compared to SAC. Our current results on our proposed benchmark are in line with previous results. We observe that both our hierarchical approach and single-level TD-MPC2 solves humanoid control tasks in <3M environment steps, whereas SAC and DreamerV3 fail to learn within 3M steps.

---

> ### Author Response · Authors · 2024-11-27
> **Added new baseline**
>
> As promised, we would like to update the reviewer with preliminary results for the baseline that consists of a high-level DreamerV3 w/ a low-level TD-MPC2 agent (pretrained as in our method). We run this baseline on three tasks: stand, walk, run with 10 random seeds per task. Results are shown [on this link](https://i.imgur.com/dpPG7fq.png) as well as in Figure 5 of our revised manuscript. We have results for up to 1.7M environment steps at the moment (56 hours of training) and expect it to take another 3 days for training to complete. However, we believe that the conclusion is clear: a TD-MPC2 backbone (which uses planning) is crucial to performance in whole-body humanoid control, at every level of the (hierarchical) architecture. We summarize our experimental results wrt this particular contribution as follows:
>
> - Single-level SAC and single-level DreamerV3 achieves no meaningful performance on our tasks
> - Single-level TD-MPC2 achieves good data-efficiency and asymptotic performance but produces highly unnatural motions
> - High-level SAC + low-level TD-MPC2 achieves no meaningful performance
> - High-level DreamerV3 + low-level TD-MPC2 achieves non-trivial performance on a limited number of tasks and random seeds but is unstable and prone to divergence
> - Our method, Puppeteer, achieves strong data-efficiency, asymptotic performance, *and* produces natural and human-like motions
> - Disabling planning at any level of our hierarchical approach degrades performance significantly (Figure 8)
>
> We hope that these additional baselines (high-level SAC / DreamerV3 + low-level TD-MPC2) address the reviewer's concerns.
>
> ----
>
> **Edit:** We have updated the manuscript + link shared here with current "DreamerV3 w/ low-level TD-MPC2" baseline results one last time before paper revision closes. We believe that performance for this baseline is unlikely to improve with additional training but will run it to completion.

---

> ### Comment · Reviewer_2bf6 · 2024-11-28
> **looking forward to the results**
>
> Thank you for working hard on implementing some of the proposed baselines. I strongly believe that they are needed to clearly understand your contribution. The results so far are looking very promising and I'm looking forward to the final results. Can you please elaborate on the experimental setup for this comparison. Mainly, I'm interested in how much effort was put into hyper-parameter tuning of the baselines in comparison to your method (especially when it comes to things like horizon length, latent space dimensionality/ general network architecture, batch size and learning rates).

---

> > ### Author Response · Authors · 2024-11-28
> > **Re: hyper-parameters**
> >
> > We are pleased to hear that these additional baselines seemingly address your remaining concerns. We are more than happy to elaborate on the experimental setup.
> >
> > DreamerV3 and TD-MPC2 are, by design, robust to choice of hyper-parameters and do not require tuning for individual domains/tasks. These are excerpts from their respective abstracts:
> >
> > DreamerV3:
> > > Robustness techniques based on normalization, balancing, and transformations enable stable learning across domains. [...] Our work allows solving challenging control problems without extensive experimentation, making reinforcement learning broadly applicable.
> >
> > TD-MPC2:
> > > We demonstrate that TD-MPC2 improves significantly over baselines across 104 online RL tasks spanning 4 diverse task domains, achieving consistently strong results with a single set of hyperparameters.
> >
> > We briefly discuss this in Section 4.1 Experimental Details L319-324 of our paper:
> >
> > > Both our method and baselines use the same hyperparameters across all tasks, as TD-MPC2 and DreamerV3 have been shown to be robust to hyperparameters across task suites (Hansen et al., 2024; Hafner et al., 2023; Sferrazza et al., 2024). For a fair comparison, we experiment with various design choices and hyperparameter configurations for SAC and report the best results that we obtained. We provide further implementation details in Appendix D.
> >
> > which we elaborate on in Appendix D:
> >
> > > **Puppeteer.** We base our implementation off of TD-MPC2 and use default design choices and hyperparameters whenever possible. We experimented with alternative hyperparameters but did not
> > observe any benefit in doing so.
> >
> > > **TD-MPC2.** We use the official implementation available at https://github.com/nicklashansen/tdmpc2, but modify the implementation to support multi-modal observations and termination conditions as discussed in Section 3.
> >
> > > **DreamerV3.** We use the official implementation available at https://github.com/danijar/dreamerv3, and use the default hyperparameters recommended for proprioceptive DMControl tasks. A key selling point of DreamerV3 is its robustness to hyperparameters across tasks (relative to SAC), but we find that DreamerV3 does not achieve any non-trivial performance
> > on our task suite. While DreamerV3 is a model-based algorithm, it does not use planning, which the ablation in Figure 8 (hierarchical planning) finds to be a key driver of performance in Puppeteer and TD-MPC2.
> >
> > > **SAC.** We benchmark against the implementation from https://github.com/denisyarats/pytorch_sac (Yarats & Kostrikov, 2020) due to its strong performance on lower-dimensional DMControl tasks as well as its popularity among the community. We modify
> > the implementation to support early termination. We experiment with a variety of design choices
> > and hyperparameters as we find vanilla SAC to suffer from numerical instabilities on our task suite
> > (presumably due to high-dimensional observation and action spaces), but are unable to achieve
> > non-trivial performance. [...] Design choices and hyperparameters that we experimented with are
> > as follows:
> > | **Design choice**     | **Values**                   |
> > |-----------------------|------------------------------|
> > | Number of Q-functions | 2,5                          |
> > | TD-target             | Default, REDQ (Chen et al., 2021)   |
> > | Activation            | ReLU, Mish, LayerNorm + Mish |
> > | MLP dim               | 256, 512, 1024               |
> > | Batch size            | 256, 512                     |
> > | Learning rate         | 3 × 10−4, 1 × 10−3           |
> >
> > We use the same hyperparameters and experimental setup in the hierarchical versions of SAC and DreamerV3 as in the single-level versions.
> >
> > We hope that this clears up any confusion regarding the experimental setup.

---

> > > ### Author Response · Authors · 2024-11-29
> > > **Final baseline results**
> > >
> > > Dear reviewer 2bf6,
> > >
> > > Thanks again for all the constructive feedback and discussion so far.
> > >
> > > All 30 runs of the new DreamerV3 w/ low-level TD-MPC2 baseline have now completed. The complete results are available [on this link](https://i.imgur.com/ShPGaJo.png). We believe that our conclusions summarized in [our previous comment](https://openreview.net/forum?id=7wuJMvK639&noteId=IH7pBIiY8D) still hold true: *a TD-MPC2 backbone (which uses planning) is crucial to performance in whole-body humanoid control, at every level of the (hierarchical) architecture.*
> > >
> > > Based on our logged training metrics, the instability of DreamerV3 on the *stand* and *walk* tasks appears to be due to divergence of the policy and critic networks of DreamerV3. This is a common phenomenon with high-dimensional continuous action spaces, and SAC is equally prone to such instabilities. Looking at individual seeds on the *stand* task, 6 seeds of DreamerV3 briefly experience signs of learning followed by training divergence, and the remaining 4 seeds fail to learn at all. The original DreamerV3 paper primarily considers tasks with discrete action spaces for which this is often less of an issue.
> > >
> > > We hope that these two additional baselines, SAC w/ low-level TD-MPC2 and DreamerV3 w/ low-level TD-MPC2, address the reviewer's concerns, and that the reviewer would be willing to reconsider their initial score as a result.

---

> > > > ### Author Response · Authors · 2024-12-02
> > > > **Discussion ends TODAY**
> > > >
> > > > Dear reviewer,
> > > >
> > > > This is a friendly reminder that the discussion period is ending **TODAY**, December 2. We believe that our responses and additional experimental results address the reviewer's concerns. Since you previously stated that you would be willing to increase your score if we included these additional baselines, we would really appreciate it if you could take a moment to reevaluate our revised manuscript (including the baselines that you suggested along with changes suggested by other reviewers) and consider updating your score accordingly. Thanks again for your time and valuable feedback!
> > > >
> > > > Best,
> > > >
> > > > Authors of Puppeteer

---

> > > > > ### Comment · Area_Chair_kWAX · 2024-12-02
> > > > >
> > > > > As the authors point out, the discussion period ends today. Please be sure to read their most recent posts, which address your questions about the experimental setup.
> > > > >
> > > > > Best,\
> > > > > AC

---

> > > > > > ### Author Response · Authors · 2024-12-03
> > > > > > **Final reminder**
> > > > > >
> > > > > > Dear reviewer 2bf6,
> > > > > >
> > > > > > This is a **final reminder** that the discussion period ends in approx. **6 hours** (Dec 2, anywhere on earth). Since we included the additional baselines requested by the reviewer in our previous response, and the reviewer appears to have no other concerns, we would really appreciate it if they would be willing to acknowledge our response and update their score accordingly.
> > > > > >
> > > > > > Best,
> > > > > >
> > > > > > Authors of Puppeteer

---

### Official Review · Reviewer_C5cF · 2024-11-04

**Soundness:** 3
**Presentation:** 3
**Contribution:** 3
**Rating:** 8
**Confidence:** 3

**Summary:**

This paper explores the high-dimensional humanoid control from visual observations. Specifically, the proposed approach is based on two RL-trained agent models.  High-level agent generates reference trajectories from visual observations. Low-level agent focuses on tracking these trajectories using current low-dimensional state information. The proposed method demonstrated enhanced natural motion control of a 56-DoF simulated humanoid, outperforming baseline models according to experimental results and a user study.

**Strengths:**

1. The research addresses a significant and practical challenge in generalist agents: controlling a humanoid agent from visual observations using generalizable world models.
2. The methodology involves training a low-level agent on trajectory tracking that is adaptable across a range of control tasks, showing promising generalizability.
3. A high-level agent controls the humanoid from visual observations, a task-specific but broadly applicable approach in real-world scenarios.
4. A user study validates that the proposed method enables more natural humanoid control, which is preferred by participants.

**Weaknesses:**

1. The evaluation heavily relies on the "naturalness" of movements, which depends on subjective human judgments of what is considered "human-like." This criterion, while important, may not fully evaluate the feasibility of such motions in actual humanoid robots, which face different kinematic and dynamic constraints than humans.
2. Based on Figure 5, the episodic return of the baseline TD-MPC2 is comparable or superior to the proposed method across most tasks. It would be beneficial to evaluate other performance metrics such as survival rates or survival times on the final-trained model to provide a more comprehensive evaluation.
3. The paper claims "Zero-shot generalization to larger gap lengths," yet does not compare these results with baseline methods. Including comparative generalization data for the baseline TD-MPC method would strengthen claims of superior generalization.
4. Minor issue:
a) Resource Efficiency: The two level agents training approach might require significantly more time and resources than single-agent baselines. Comparing memory usage, training duration, and inference times across methods would provide critical insights into the practicality of the proposed method.
b) Model Reusability: The low-level tracking agent is described as reusable across tasks but it is unclear if this model is applicable only to 56-DoF humanoids or if it can be adapted to different control dimensions.
c) There is a typo in the Problem Formulation section (page 2). The environment transition function should be denoted as L, not S.

**Questions:**

1. How relevant is the metric of "naturalness" in real-world humanoid control, and is it sufficient to evaluate humanoid trajectory tracking effectively?
2. It would be beneficial to include a comparative study on the survival rate or survival time when using the final-trained model
3. It would be helpful to include baseline experiments focused on "zero-shot generalization."
4. Please provide details on memory usage, training time, and control (or inference) time across different methods.
5. Is the low-level tracking effectively transferred to control different humanoid models with varying degrees of freedom?

---

> ### Author Response · Authors · 2024-11-22
> **Thank you!**
>
> We thank the reviewer for their valuable feedback. We address your comments in the following.
>
> ----
>
> **Q:** How relevant is the metric of "naturalness" in real-world humanoid control, and is it sufficient to evaluate humanoid trajectory tracking effectively?
>
> **A:** This is a great question. We do believe that policies that behave “naturally” and “human-like” are inherently valuable. For example, the physical safety of a robot and its surroundings is often a priority in robotics applications, which is not unlike when humans perform everyday tasks. It is desirable for humans and robots alike to complete tasks in ways that are efficient yet reliable and safe, and humans do tend to prefer when other agents (e.g.) behave in a predictable manner. This preference is, to some extent, validated in our user study. Biasing an RL policy towards human MoCap data is an effective way to embed behavioral preferences in a humanoid setting. Preferences can in principle be specified via reward engineering, but designing a reward that accurately conveys specific preferences can be very non-trivial compared to a more data-driven approach like ours. We focus on humanoids in this work, but our approach could in principle be applied to other embodiments for which a dataset of reference motions are available. At a high level, you can think of our problem setting as conceptually similar to the alignment problem in e.g. LLMs.
>
> ----
>
> **Q:** It would be beneficial to include a comparative study on the survival rate or survival time when using the final-trained model
>
> **A:** Agreed! Table 1 of our manuscript already compares gait + average episode length (survival time) over the course of training. Following your suggestion, we now report episode length at 1M environment steps + at convergence. We generally observe that TD-MPC2 achieves similar episode return and survival times as Puppeteer at convergence, but that Puppeteer has significantly higher survival times both throughout training and at the 1M snapshot. We conjecture that the behavioral prior of the low-level tracking agent is especially helpful in the early stages of training.
>
> |           | Eplen @ 1M ↑ | Eplen (final) ↑ | Torso height ↑ |
> |-----------|--------------|-----------------|----------------|
> | TD-MPC2   | 66.9 +- 9.8  | 181.6 +- 28.1   | 85.9 +- 4.7    |
> | Puppeteer | **115.9 +- 5.2** | 159.3 +- 5.9    | **96.0 +- 0.2**    |
>
> We **bold** numbers that are at least one std.dev. apart. We have updated Table 1 with these new metrics in our revised manuscript.
>
> ----
>
> **Q:** It would be helpful to include baseline experiments focused on "zero-shot generalization."
>
> **A:** Great suggestion! We have added baseline results to our zero-shot generalization experiments in Figure 9 of our updated manuscript. The figure is also viewable here [https://i.imgur.com/gP0MLW0.png](https://i.imgur.com/gP0MLW0.png) for your convenience. We observe that Puppeteer generally is more robust to changes in gap length than the TD-MPC2 baseline, which we attribute to its more stable gait. Videos of our method performing this task with varying gap lengths are shown on our project webpage.
>
> ----
>
> **Q:** Please provide details on memory usage, training time, and control (or inference) time across different methods.
>
> **A:** Thank you for the suggestion! We already provide training times and hardware requirements of Puppeteer in Section 4.1 of our manuscript, but agree that a more comprehensive comparison to baselines would be informative. We have added a comparison of wall-time, inference time, and GPU memory for Puppeteer, TD-MPC2, SAC, and DreamerV3. Overall, wall-time and system requirements of Puppeteer are mostly comparable to that of TD-MPC2 across both state-based and visual RL. SAC runs approx. 3.6x faster than Puppeteer, but does not achieve any meaningful performance on our benchmark; DreamerV3 runs approx. 2.3x **slower** than Puppeteer and also does not achieve any meaningful performance.
>
> |           | Wall-time (h / 1M steps) | Inference time (ms / step) | GPU memory (GB) |
> |-----------|--------------------------|----------------------------|-----------------|
> | SAC       | 5.9 (vision: N/A)        | 2.2                        | 0.4             |
> | DreamerV3 | 50.2 (vision: N/A  |  18.7                     | 3.9             |
> | TD-MPC2   | 18.6 (vision: 25.2)      | 50.8                       | 0.5             |
> | Puppeteer | 21.8 (vision: 29.0)      | 88.2                       | 0.6             |
>
> These results are now included in Appendix C of our updated manuscript, along with a detailed description of how the numbers are obtained.
>
> **Edit:** Added DreamerV3 numbers to the table above.
>
> ----
>
> (1/2)

---

> > ### Author Response · Authors · 2024-11-22
> > **Thank you! Part 2**
> >
> > **Q:** Is the low-level tracking effectively transferred to control different humanoid models with varying degrees of freedom?
> >
> > **A:** The low-level tracking agent takes current joint positions as input and outputs delta joint positions for each of the 56 actuated joints in our humanoid model, so we doubt that a trained agent would be able to control a different humanoid model without finetuning. However, we do believe that our tracking agent would be able to control a variety of different embodiments when either trained from scratch on the new embodiment or with some amount of finetuning. Given that the hierarchical interface is 3D end-effector positions, it is possible that the high-level agent would be able to control a new humanoid model without any additional training, as long as a new low-level tracking agent is obtained for that model. This would definitely be an interesting direction for future research.
> >
> > ----
> >
> > We believe that our response addresses your main concerns. However, if that is not the case please let us know if you have any additional comments. We would greatly appreciate it if the reviewer could provide us with precise and actionable feedback such that we can fully address your concerns. Thanks again for your time and valuable feedback!
> >
> > (2/2)

---

> > > ### Author Response · Authors · 2024-11-24
> > > **A gentle reminder**
> > >
> > > Dear reviewer C5cF,
> > >
> > > The end of the discussion period is rapidly approaching, and we would really appreciate it if you had a moment to look over our response and changes to the manuscript. We believe that your main concerns are addressed by our response, but if not we will be more than happy to work with you on further addressing them. Thanks again for your time and for serving as a reviewer!

---

> > > > ### Author Response · Authors · 2024-11-25
> > > >
> > > > We have now updated the Table above with wall-time, inference time, and GPU memory requirements of DreamerV3 on our benchmark, as previously promised. We would greatly appreciate it if the reviewer could take a moment to review our response and changes to the manuscript.

---

> > > > > ### Author Response · Authors · 2024-11-27
> > > > > **Reminder**
> > > > >
> > > > > Thanks again for your thoughtful review and feedback so far. As the rebuttal period is coming to a close, we would like to encourage further discussion or clarification on any remaining points. We are happy to address any concerns to ensure all perspectives are thoroughly considered.

---

> > > > > > ### Author Response · Authors · 2024-12-01
> > > > > > **Discussion period is ending**
> > > > > >
> > > > > > Dear reviewer C5cF,
> > > > > >
> > > > > > As the discussion period concludes on December 2 (in **2** days!) and we have not heard from you yet, we wanted to kindly remind you to read and reply to our rebuttal as soon as possible. We welcome you to share any remaining concerns or feedback. If our responses have adequately addressed your comments, we would greatly appreciate it if you could update your score accordingly.

---

> > > > > > > ### Author Response · Authors · 2024-12-02
> > > > > > > **Discussion ends TODAY**
> > > > > > >
> > > > > > > Dear reviewer,
> > > > > > >
> > > > > > > This is a final reminder that the discussion period is ending **TODAY**, December 2, and we still have not heard from you. We would really appreciate it if you could take a moment to go through our response + revised manuscript, and consider increasing your score if our rebuttal has addressed your concerns. Thanks again for your time and valuable feedback.
> > > > > > >
> > > > > > > Best,
> > > > > > >
> > > > > > > Authors of Puppeteer

---

> > > > > > > > ### Comment · Reviewer_C5cF · 2024-12-02
> > > > > > > > **Reviewer Comment**
> > > > > > > >
> > > > > > > > Thanks for adding additional results to address the reviewers' questions. All my concerns are addressed. I am raising my score accordingly.

---

> > > > > > > > > ### Author Response · Authors · 2024-12-02
> > > > > > > > > **Thank you**
> > > > > > > > >
> > > > > > > > > Thank you for acknowledging our response! We are pleased to hear that our changes have addressed your concerns and that you are willing to raise your score as a result.

---

### Official Review · Reviewer_HShf · 2024-11-05

**Soundness:** 3
**Presentation:** 3
**Contribution:** 2
**Rating:** 8
**Confidence:** 4

**Summary:**

The paper proposes a hierarchical world model for whole-body humanoid control based on RL. The framework separates high-level and low-level control, with a high-level puppeteering agent providing commands for a pre-trained low-level tracking agent, which executes detailed joint movements. Key contributions include a task suite for visual humanoid control, a hierarchical control model using RL without pre-defined reward designs, metrics for "naturalness" in motion, and thorough analysis through ablation studies and user preference tests.

**Strengths:**

1. The hierarchical world model, which integrates high-level visual guidance with low-level proprioceptive control, is novel in its simplicity and efficacy, especially in achieving natural motion without predefined rewards or skill primitives.

2. Puppeteer advances visual whole-body humanoid control by setting new standards for naturalness and efficiency in motion synthesis. The zero-shot generalization to unseen tasks demonstrates the model’s potential for practical application.

**Weaknesses:**

1. Lack of low-level tracking performance evaluation. There is no evaluation or metrics for the tracking accuracy of success rate. There are several works both from simulated avatars community [1,2] and real-world humanoids [3,4] that evaluate the tracking performance. I am supurised that these works are not mentioned and their metircs are not used for evaluation in this work.


[1] Luo, Z., Cao, J., Kitani, K., & Xu, W. (2023). Perpetual humanoid control for real-time simulated avatars. In Proceedings of the IEEE/CVF International Conference on Computer Vision (pp. 10895-10904).

[2] Won, J., Gopinath, D., & Hodgins, J. (2022). Physics-based character controllers using conditional vaes. ACM Transactions on Graphics (TOG), 41(4), 1-12.

[3] Cheng, X., Ji, Y., Chen, J., Yang, R., Yang, G., & Wang, X. (2024). Expressive whole-body control for humanoid robots. arXiv preprint arXiv:2402.16796.

[4] He, T., Luo, Z., Xiao, W., Zhang, C., Kitani, K., Liu, C., & Shi, G. (2024). Learning human-to-humanoid real-time whole-body teleoperation. arXiv preprint arXiv:2403.04436.


2. The lack of interface design discuss. The paper proposes to use high-level controller to generate positions of tracking keypoints. This might be one way for reuse the low-level skills for downstream tasks, but there are many existing designs in prior works [5] [6] that not are compared in this work. To me, the idea of training low-level tracking policy for skills reuse is a long-standing idea, but the interface of this hierarchy matters a lot. I'd love to see more comparison on this.

[5] Tessler, C., Kasten, Y., Guo, Y., Mannor, S., Chechik, G., & Peng, X. B. (2023, July). Calm: Conditional adversarial latent models for directable virtual characters. In ACM SIGGRAPH 2023 Conference Proceedings (pp. 1-9).

[6] Luo, Z., Cao, J., Merel, J., Winkler, A., Huang, J., Kitani, K., & Xu, W. (2023). Universal humanoid motion representations for physics-based control. arXiv preprint arXiv:2310.04582.

3. The source of naturalness is unclear. The low-level tracking policy might be conditioned on human motion pirors, but if the tracking policy is good enough, it should be able to produce what TD-MPC2 achieves. Also, if the advantage of this paper is sample-efficiency and naturalness, a key baseline here would TD-MPC2+ AMP, which is missing. That being said, all the experimental results make sense to me, but the key comparison experiments are missing somehow.

**Questions:**

All my questions are listed in the weakness part.

---

> ### Author Response · Authors · 2024-11-22
> **Thank you!**
>
> We thank the reviewer for their valuable feedback. We address your comments in the following.
>
> ----
>
> **Q:** Lack of low-level tracking performance evaluation. There is no evaluation or metrics for the tracking accuracy of success rate.
>
> **A:** While we predominantly focus on the utility of the low-level tracking agent for downstream tasks, we agree that a more thorough evaluation of the tracking performance itself would be informative. We include 4 new evaluations: (1) a collection of tracking visualizations for readers to qualitatively evaluate our approach, (2) success rate as defined by [1] Luo, Z., Cao, J., Kitani, K., & Xu, W. (2023), (3) average tracking error, and (4) CoMic [2] score across all 836 clips. Results are shown below, and have also been added to our revised manuscript along with more precise definitions of the metrics used.
>
> |              | Success rate (%) ↑ | Tracking error ↓ | CoMic score ↑ |
> |--------------|--------------------|------------------|---------------|
> | Offline only | 6.2                | 0.503            | 42.6          |
> | 5% data      | 74.4               | 0.260            | 45.4          |
> | 25% data     | 79.6               | 0.225            | 46.3          |
> | 75% data     | 87.9               | 0.202            | 47.4          |
> | Ours         | **88.3**               | **0.187**            | **48.7**          |
>
> We also appreciate the additional references; they are now cited in the related work section of our updated manuscript. Thanks again for your suggestions!
>
> [2] Hasenclever, L., Pardo, F., Hadsell, R., Heess. N., Merel, J., “CoMic: Complementary Task Learning & Mimicry for Reusable Skills”, 37th International Conference on Machine Learning (2020)
>
> ----
>
> **Q:** The lack of interface design discuss. [...] To me, the idea of training low-level tracking policy for skills reuse is a long-standing idea, but the interface of this hierarchy matters a lot. I'd love to see more comparison on this.
>
> **A:** We agree that the interface between hierarchies is important, and that including a comparison to e.g. an interface based on latent actions (as opposed to our explicit 3D end-effector joint commands) would be informative. We experimented with such an approach in the early stages of our project but found our current approach to be more reliable for whole-body humanoid control. Including such a comparison will take longer than the span of this rebuttal period, but we are committed to adding it for the camera-ready version. Would that address your concern?
>
> ----
>
> **Q:** The source of naturalness is unclear. The low-level tracking policy might be conditioned on human motion pirors, but if the tracking policy is good enough, it should be able to produce what TD-MPC2 achieves. Also, if the advantage of this paper is sample-efficiency and naturalness, a key baseline here would TD-MPC2+ AMP, which is missing.
>
> **A:** We agree that further exploring how the data mixture and design choices affect generated motions would be really interesting and informative. If the reviewer believes that this would add significant value to the manuscript, we will be more than happy to add a TD-MPC2+AMP baseline to the camera-ready version, along with an additional ablation of how different mixtures of MoCap data (e.g. excluding all “running” clips from the pretraining data) influence the behavior of learned policies. We would like to point out, however, that the algorithmic complexity of AMP (inverse RL with a learned GAN discriminator + unsupervised skill discovery) is significantly more complicated than our approach (RL without any bells and whistles + no hyperparameter-tuning), and that the original AMP work only achieved downstream task transfer in simple state-based (i.e., no visual inputs) reaching and velocity control tasks while using several orders of magnitude more environment steps to do so. For example, AMP required approx. 800M steps to learn the reaching task, whereas our method learns visuo-motor control tasks such as running in an environment with randomized obstacles in as little as 3M steps.
>
> ----
>
> We believe that our response addresses your main concerns. However, if that is not the case please let us know if you have any additional comments. We would greatly appreciate it if the reviewer could provide us with precise and actionable feedback such that we can fully address your concerns. Thanks again for your time and valuable feedback!

---

> > ### Comment · Reviewer_HShf · 2024-11-23
> > **Thank you**
> >
> > Thank you for the detailed response. The authors have addressed the majority of my concerns and I have raised my evaluation accordingly to accept.

---

> > > ### Author Response · Authors · 2024-11-24
> > > **Thanks again!**
> > >
> > > We are happy to hear that our response + changes to the manuscript addresses your concerns, and that you have decided to raise your score. Thanks again for your time and valuable feedback, we really appreciate it!

---

### Author Response · Authors · 2024-11-22
**General comment**

We thank all reviewers for their thoughtful comments, and are especially pleased to see that the reviewers agree that:
- The paper is **well written** and enjoyable to read (reviewers 2bf6, EisB)
- Our method is conceptually **simple yet generates natural motions** (reviewers HShf, EisB)
- **Integration of vision** in whole-body humanoid control is exciting (reviewers HShf, C5cf, 2bf6, EisB)
- Our proposed benchmark **tasks and experiments are interesting** (reviewers HShf, C5cF, 2bf6, EisB)
- Our choice of **baselines and ablations are informative** (reviewer 2bf6)

----

### Summary of revisions

We have revised our manuscript based on your feedback – a summary of changes is available below. These changes are highlighted (red) in the updated manuscript. We have also responded to your individual comments.

- **(HShf)** Added 4 new evaluations of the low-level tracking policy: tracking visualizations for qualitative assessment, success rate, average tracking error, and CoMic score. Results are reported in Table 2 for our method + ablations, as well as in [this](https://openreview.net/forum?id=7wuJMvK639&noteId=y6ra8KpCFC) reply to reviewer HShf.
- **(C5cf)** Added a new survival time (episode length) metric in downstream task evaluations, reported at 1M steps and at convergence. Results have been added to Table 1, as well as in [this](https://openreview.net/forum?id=7wuJMvK639&noteId=JLhyohwGJT) reply to reviewer C5cf.
- **(C5cf)** Added a table with wall-time, inference time, and GPU memory requirements for our method and baselines across both state-based and visual RL tasks from our benchmark. Results are reported in Table 3, as well as in [this](https://openreview.net/forum?id=7wuJMvK639&noteId=JLhyohwGJT) reply to reviewer C5cf.
- **(C5cf)** Added baseline results for our zero-shot generalization experiments in Figure 9. We also share the figure here https://i.imgur.com/gP0MLW0.png for your convenience.
- **(2bf6)** Added a new baseline that uses the same low-level agent (based on TD-MPC2) as our method but replaces the high-level agent with a **SAC** policy. Results are reported in Figure 5, as well as here https://i.imgur.com/ShPGaJo.png for your convenience.
- **(2bf6)** Added a new baseline that uses the same low-level agent (based on TD-MPC2) as our method but replaces the high-level agent with a **DreamerV3** policy. Results are reported in Figure 5, as well as here https://i.imgur.com/ShPGaJo.png for your convenience.
- **(2bf6)** Updated Section 3.2 to clarify that the high-level policy acts at a fixed frequency regardless of whether previous commands were achieved.

----

Again, we thank the reviewers for their constructive feedback. We have made a significant effort to address all comments made by reviewers, and we hope that reviewers will take our responses into consideration when deciding whether to revise their score. If any reviewer believes that their concerns have not been addressed by our rebuttal, we would be very happy to work with them to address any further comments.

Thank you for serving as a reviewer!


Best,

Authors of Puppeteer

---

### Meta-Review · Area_Chair_kWAX · 2024-12-23

**Metareview:**

The paper proposes Puppeteer, a hierarchical reinforcement learning (RL)-based framework for whole-body control of humanoid robots based upon visual observations. The high-level model generates reference commands for a pre-trained low-level policy responsible for trajectory tracking. The experimental analysis emphasizes the "naturalness" of the resulting motion, with performance gains over contemporary baselines.

The paper was reviewed by four referees, who despite attempts by the AC, were unable to come to a consensus on the merits of the paper. Two reviewers (HShf and C5cF) appreciated the broader advancements that the paper provides to humanoid control. Reviewers 2bf6 and EisB found that it is well written and a pleasure to read. Reviewers HShf and EisB appreciated Puppeteer's hierarchical approach, but disagreed on its novelty. Meanwhile, at least two reviewers (C5cf and 2bf6) noted that the results show that the TD-MPC2 baseline is comparable to if not better than the proposed method. Related, several reviewers emphasized the fact that the evaluation focuses on the "naturalness" of the resulting motion, however "naturalness" is subjective and does not afford a standard metric. In their response, the authors note the importance of realizing motions that are natural, which the AC agrees with, but they did not address its subjectivity. Related, Reviewer 2bf6 questions the role of hierarchy in ensuring the naturalness of the resulting behavior, which the authors attribute to pretraining the motions on a large-scale collection of human motion capture data. Additionally, some reviewers identified the need for comparisons to other baselines, while others requested and evaluation of low-level tracking performance, which the authors made an effort to address during the rebuttal period by including additional results.

The AC recognizes the importance of realizing humanoid motions that are natural and human-like, particularly for settings where robots are expected to interact with people. That said, the paper would benefit from a more concrete discussion of what novel aspects of the method are integral to realizing natural behaviors. THe significance of Puppeteer's contributions in this regard is not clear, particularly in light of the authors' claim that it is due to having pretrained on a large amount of human motion capture data, which one can argue constitutes an incremental contribution.

**Additional Comments On Reviewer Discussion:**

The AC recognizes that some of the reviewers did not participate in the discussion period as soon as one would like. Each reviewer ended up responding to the authors' rebuttal, some perhaps at the request of the AC. This is is in slight conflict with the authors' claims that Reviewers 2bf6 and EisB were not active in the discussion. While they were not as active as some may have desired, the did respond before the end of the discussion period.

Meanwhile, the AC urged the reviewers to try and come to a consensus on the paper. Unfortunately, they remained split on the paper's merits.

---

### Decision · Program_Chairs · 2025-01-22

Accept (Poster)